# ATTENTION LAYERS ADD INTO LOW-DIMENSIONAL RESIDUAL SUBSPACES

Attention outputs exhibit pronounced low-rank structure compared to residual streams and MLP outputs, indicating that the attention layer writes into a subspace of the residual stream.

Low effective dimensionality of activations is a root cause of dead features in sparse dictionary learning methods. Setting feature directions in the active subspace mitigates this issue.

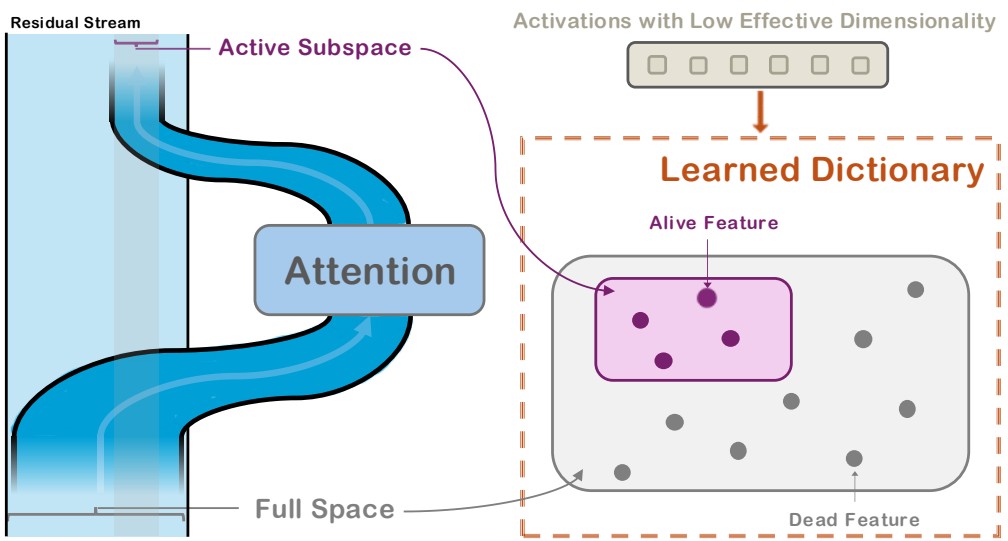

## ABSTRACT

Transformer architectures, and their attention mechanisms in particular, form the foundation of modern large language models. While transformer models are widely believed to operate in high-dimensional hidden spaces, we show that attention outputs are confined to a surprisingly low-dimensional subspace, where about 60% of the directions account for 99% of the variance–a phenomenon that is consistently observed across diverse model families and datasets, and is structurally imposed by the attention output projection matrix. Critically, we find this low-rank structure as a key factor of the prevalent dead feature problem in sparse dictionary learning, where it creates a mismatch between randomly initialized features and the intrinsic geometry of the activation space. Building on this insight, we propose a subspace-constrained training method for sparse autoencoders (SAEs), initializing feature directions into the active subspace of activations. Our approach reduces dead features from 87% to below 1% in Attention Output SAEs with 1M features, and can extend to other sparse dictionary learning methods. Our findings provide both new insights into the geometry of attention and practical tools for improving sparse dictionary learning in large language models. Code is available at https://anonymous.4open.science/r/Language-Model-SAEs-2B1D/.

# 1 INTRODUCTION

Over the past years, mechanistic interpretability has shifted from a collection of proof-of-concept tools (Olsson et al., 2022; Wang et al., 2022; Meng et al., 2023; Gould et al., 2023) toward a fast-growing, scale-driven field (Ameisen et al., 2025; Lindsey et al., 2025). This transformation is driven by a wave of sparse dictionary learning methods–such as sparse autoencoders (SAEs) and their variants (Cunningham et al., 2023; Bricken et al., 2023b; Lindsey et al., 2024b), transcoders (Dunefsky et al., 2024; Ge et al., 2024), and low-rank sparse attention (He et al., 2025)–that once targeted small models but are now being pushed to larger architectures and wider model families (Templeton et al., 2024; Gao et al., 2024; Hazra et al., 2025). As these approaches scale in performance and model coverage, they provide increasingly complete and fine-grained explanations of neural network behavior (Lindsey et al., 2024a; Gao et al., 2024).

However, scaling these approaches presents practical difficulties (Templeton et al., 2024; Gao et al., 2024; Mudide et al., 2025). As models and feature dictionaries grow, the number of parameters increases rapidly, driving up computational costs. At the same time, the prevalence of dead features leads to substantial waste in computation and memory (Templeton et al., 2024; Kissane et al., 2024), limiting the efficiency of interpretability methods. In this work, we identify **low-rank activation structure as a major driver of dead features** (Section 5.1).

In Section 4, we show that **attention outputs exhibit a remarkably strong low-rank structure** compared to multilayer perceptron (MLP) outputs and residual streams. Through singular value decomposition and effective dimensionality analyses (Roy & Vetterli, 2007; Staats et al., 2025), we demonstrate that this phenomenon holds universally across layers, datasets, and model families—Llama 3.1 (Dubey et al., 2024), Gemma 2 (Rivière et al., 2024), and Qwen 3 (Yang et al., 2025), which is consistent with the universality hypothesis (Olah et al., 2020; Chughtai et al., 2023; Gurnee et al., 2024; Wang et al., 2025). We further trace the origin of this low-rank structure to the anisotropy of the output projection matrix $W^O$, which compresses the multi-head outputs into a lower-dimensional subspace.

In Section 5, we investigate how the low-rank nature of attention outputs interacts with SAE training. By evaluating the full suite of open-source SAEs from *LlamaScope* (He et al., 2024), we show that low effective dimensionality strongly correlates with the number of dead features, suggesting a mismatch between random initialization and the low-dimensional geometry of the activations. Drawing inspiration from Phan et al. (2025)'s principal component initialization for the first network layer, we propose *Active Subspace Initialization*, which aligns SAE features with the active subspace of activations, **substantially reducing dead features while improving reconstruction**. Following Lindsey et al. (2024a) and Gao et al. (2024), we conduct scaling experiments, which further reveal that ASI achieves superior reconstruction across feature counts, and when combined with SparseAdam[1], it achieves the best reconstruction in large scale and reduces dead features from 87% to below 1% in Attention Output SAEs with 1M features trained on Llama-3.1-8B (Dubey et al., 2024).

Furthermore, we show that **Active Subspace Init can generalize to sparse replacement models** (He et al., 2025; Dunefsky et al., 2024; Ameisen et al., 2025) (Section 5.4). When applied to other sparse dictionary learning methods, our initialization procedure systematically reduces the prevalence of dead parameters across architectures.

# 2 RELATED WORK

## 2.1 LOW-RANKNESS AND LINEAR SUBSPACE STRUCTURE IN NEURAL REPRESENTATIONS

A long line of research has shown that neural representations frequently concentrate in low-dimensional linear subspaces, forming structured semantic or bias subspaces in embedding spaces (Mikolov et al., 2013; Pennington et al., 2014; Bolukbasi et al., 2016; Vargas & Cotterell, 2020). These studies reveal that low-dimensional linear structure is a broader phenomenon of neural representations. More recent work further cautions that such subspaces may sometimes be

---

[1]https://docs.pytorch.org/docs/stable/generated/torch.optim.SparseAdam.html

misleading or illusory (Makelov et al., 2024), and understanding their reliability remains an open challenge (Sharkey et al., 2025).

Within attention mechanisms specifically, prior work has investigated various notions of "low-rankness": low-rank approximation of attention patterns (Wang et al., 2020; Tay et al., 2020; Raganato et al., 2020), low-rank parameterization for model compression (Noach & Goldberg, 2020; Hu et al., 2022), and the inherent low-rank bottleneck in single-head outputs (Bhojanapalli et al., 2020).

Different from these prior lines of work, we demonstrate that the multi-head self-attention outputs exhibit a low-rank structure, revealing a distinct and under-explored phenomenon.

### 2.2 SUPERPOSITION HYPOTHESIS AND SPARSE DICTIONARY LEARNING METHODS

The superposition hypothesis posits that neurons encode multiple non-orthogonal underlying features (Arora et al., 2018; Olah et al., 2020; Elhage et al., 2022; Park et al., 2024). Motivated by this view, a variety of sparse dictionary learning methods have been developed for interpretability, including sparse autoencoders and their variants (Cunningham et al., 2023; Bricken et al., 2023b; Lindsey et al., 2024b), transcoders (Dunefsky et al., 2024; Ge et al., 2024), and low-rank sparse attention (He et al., 2025). These approaches decompose activations into sparse combinations of learned features while differing in their mechanisms for predicting or approximating feature activations. Their successful application across a wide range of model scales (Templeton et al., 2024; Lieberum et al., 2024; He et al., 2024), architectures (Wang et al., 2025), and modalities (Abdulaal et al., 2024) highlights their practical effectiveness for interpretability; however, they do not constitute direct hypothesis tests of the superposition hypothesis, which remain active topics of debate (Sharkey et al., 2025).

### 2.3 DEAD FEATURES IN SPARSE DICTIONARY LEARNING METHODS

A persistent challenge in sparse dictionary learning methods is the emergence of *dead features*[2] (Templeton et al., 2024; Kissane et al., 2024), which are also referred to as *dead units* in sparse replacement models (Dunefsky et al., 2024; Ge et al., 2024; He et al., 2025). These features contribute nothing to reconstruction quality, wasting parameters and computation. Existing approaches to mitigate this issue rely on auxiliary loss terms (Gao et al., 2024; Conerly et al., 2025) or resampling strategies (Bricken et al., 2023b) to encourage feature usage.

### 2.4 PCA-INSPIRED NETWORK INITIALIZATION

A common practice applies PCA to input data for dimensionality reduction before network training (Hastie et al., 2009; Montavon et al., 2012; Jolliffe, 1986; Bishop & Nasrabadi, 2007). Recently, Phan et al. (2025) proposed *PCsInit*, which initializes the first layer weights of networks with top principal components of data—embedding the PCA transform directly into the network. This provides the model with a superior parameter set (Gu et al., 2025), boosting performance by construction.

## 3 PRELIMINARIES

### 3.1 MULTI-HEAD SELF-ATTENTION AND NOTATIONS

We consider a Transformer block with multi-head self-attention (MHSA) (Vaswani et al., 2017). Given input activations $X \in \mathbb{R}^{n \times d}$, where $n$ is the token count and $d$ is the model hidden size, each attention head $i$ computes:

$$Q_i = XW_i^Q, \quad K_i = XW_i^K, \quad V_i = XW_i^V, \quad W_i^Q, W_i^K, W_i^V \in \mathbb{R}^{d \times d_h},$$

where $d_h = d/H$ is the dimensionality of each head, and $H$ is the total number of heads. The attention weights and head outputs are then given by:

$$A_i = \mathrm{softmax}\left(\frac{Q_i K_i^\top}{\sqrt{d_h}}\right), \quad Z_i = A_i V_i \in \mathbb{R}^{n \times d_h}.$$

---

[2]Following Bricken et al. (2023b), we define a feature as dead if it never activates over 10 million tokens in this paper.

Let $Z = \text{Concat}[Z_1, \ldots, Z_H] \in \mathbb{R}^{n \times d}$ denote the concatenated outputs of all attention heads (Nanda & Bloom, 2022). The final attention output is obtained by applying the output projection:

$$O = ZW^O = [Z_1, Z_2, \ldots, Z_H] \begin{bmatrix} W_1^O \\ W_2^O \\ \vdots \\ W_H^O \end{bmatrix} = \sum_{i=1}^{H} Z_i W_i^O,$$

where each $W_h^O \in \mathbb{R}^{d_h \times d}$ is the corresponding submatrix of $W^O \in \mathbb{R}^{d \times d}$ associated with head $i$.

This formulation makes explicit that $O$ is the sum of the outputs from all heads, where each head produces a rank-$d_h$ output that is projected into the residual stream space through its corresponding $W_h^O$. Thus, $O$ represents the attention block's total contribution to the residual stream.

### 3.2 TopK Sparse Autoencoders

In this work, we adopt the TopK sparse autoencoder (TopK SAE) introduced by Gao et al. (2024). Unlike standard SAEs that impose an $\ell_1$ penalty, TopK SAE enforces exact sparsity by keeping only the top-$k$ activations in the latent representation for each input. Formally, given an input vector $x \in \mathbb{R}^d$, the encoder produces

$$z = \text{TopK}(W_e x + b_e),$$

where $\text{TopK}(v)$ sets to zero all but the largest $k$ entries of $v$. The decoder then reconstructs

$$\hat{x} = W_d z + b_d.$$

The model is trained to minimize the reconstruction loss, optionally augmented with an auxiliary loss to prevent dead latents:

$$\mathcal{L}_{\text{TopK-SAE}} = \|x - \hat{x}\|_2^2 + \alpha \cdot \mathcal{L}_{\text{aux}},$$

where $\mathcal{L}_{\text{aux}}$ is an optional term designed to penalize latents that never activate over a training period, and $\alpha$ balances reconstruction fidelity and latent utilization.

## 4 Low-Rank Structure of Attention Outputs

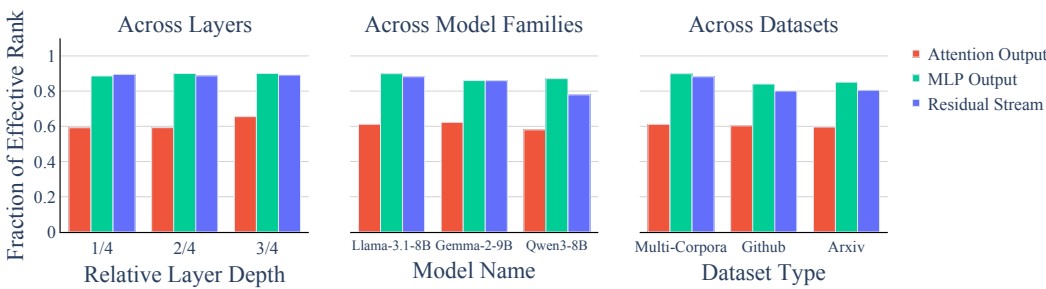

Figure 1: Across layers, model families and datasets, attention outputs exhibit dramatically lower effetive rank than residual streams and MLP outputs, indicating that the attention layer writing into a low dimensional subspace of residual stream is a universal phenomenon. Details in Section 4.1. (left) Evaluation of Llama-3.1-8B on SlimPajama (Soboleva et al., 2023) dataset. (mid) Middle-layer analysis across model families on SlimPajama dataset. (right) Middle-layer analysis of Llama-3.1-8B across datasets.

We begin by presenting our central empirical finding: in Transformer models, attention outputs consistently display the strongest low-rank structure compared to MLP outputs and residual streams. As shown in Figure 1, attention outputs have a significantly lower effective rank. This phenomenon is remarkably robust, holding across different intermediate layers, model families and datasets. These observations highlight that the attention block modifies a subspace of the residual stream, while the MLP operates nearly on the full space.

### 4.1 QUANTIFYING LOW-RANKNESS WITH RELATIVE SINGULAR VALUES

We consider the activation matrix $A \in \mathbb{R}^{n \times d}$, where each row corresponds to the activation vector of a single token. Here, $n$ denotes the number of data points and $d$ the dimensionality of the activation space (e.g., the model's hidden size). Unless otherwise specified, $\widetilde{A}$ represents mean-centered activations. Further details regarding the activation sources are provided in Appendix B.

To quantify the effective dimensionality of the data, we adopt the *effective rank* metric introduced by Roy & Vetterli (2007).

**Definition 1** (Effective Rank, Roy & Vetterli (2007)). *Let $\widetilde{A}$ be a nonzero matrix with singular value decomposition $\widetilde{A} = U\Sigma V^{\top}$, where $\Sigma = \mathrm{diag}(\sigma_1, \sigma_2, \ldots, \sigma_r)$ contains singular values in descending order. Define the normalized singular value distribution as*

$$p_k = \frac{\sigma_k}{\sum_{j=1}^{r} \sigma_j}, \quad k = 1, 2, \ldots, r.$$

*The (Shannon) entropy of this distribution is*

$$H(p_1, p_2, \ldots, p_r) = -\sum_{k=1}^{r} p_k \log p_k.$$

*Then, the effective rank of $\widetilde{A}$ is defined as*

$$\mathrm{erank}(\widetilde{A}) = \exp\big\{H(p_1, p_2, \ldots, p_r)\big\}.$$

Intuitively, the effective rank captures how evenly the singular values are distributed—higher values indicate a more isotropic spectrum, whereas lower values reflect concentration along a few dominant directions. *Fraction of effective rank* used in Figure 1 means effective rank divided by the dimension of activation space.

In addition, we compute the *fraction of downstream loss recovered* by varying the number of retained components, following Bricken et al. (2023b); Rajamanoharan et al. (2024a). Specifically, we take the language model loss with ablating the activations to zero as a baseline and report the fraction of that loss recovered as components are reintroduced. We refer readers to Appendix C for formal definition of this term.

These quantitative measures complement our core analyses by providing a numerical characterization of the low-rank structure present in activations.

### 4.2 EXPERIMENT SETTINGS

For each activation type, we collect a total of 10 million activation vectors.[3] We empirically verified this sample size suffices to ensure stable and reproducible singular spectrum analyses in Appendix D.

Unless otherwise specified, all experiments in Section 4 run on the middle layer of Llama-3.1-8B (layer 15, zero-indexed), using SlimPajama dataset.

### 4.3 EMPIRICAL EVIDENCE OF LOW-RANK STRUCTURE

We draw our findings from three lines of evidence:

**Low Effective Rank of Attention Output**    Attention outputs have a effective rank of around 60% of the total dimensionality. In contrast, MLP outputs and the residual streams show much higher effective rank around 90% (Figure 1).

**Rapid Singular Spectral Decay in Attention Output**    This is quantitatively evidenced by the number of components retaining significant energy: only 74.7% singular values exceed 1% of the maximum in attention output, versus 100.0% for MLP output and residual stream (Figure 2a).

---

[3]In rare cases, outlier activations inflate variance along certain directions, potentially biasing variance-based dimensionality estimates. To mitigate this, we exclude activations whose norms exceed $5\sigma$ from the mean.

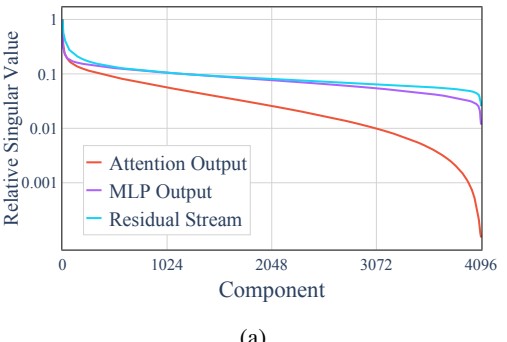 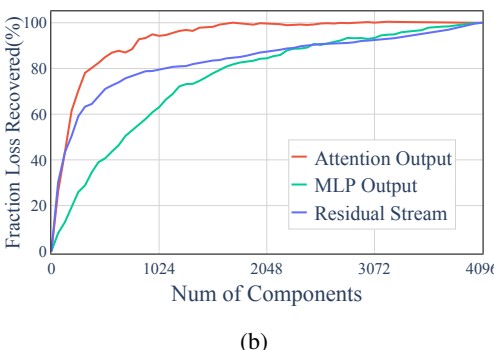

(a)                    (b)

Figure 2: **(a)** The attention output is the most low-rank, as indicated by the sharpest decay in singular values. **(b)** Fraction of loss recovered using varying numbers of top singular components.

**Efficient Downstream Loss Recovery**     Compared to zero ablation, attention output requires only 39.1% of dimensions to recover over 99% of the downstream loss, versus 95.3% and 96.9% of the dimensions for MLP outputs and residual streams to recover the same proportion (Figure 2b).

More results of these metrics across different layers, models and datasets are shown in Appendix E.

### 4.4    LOW-RANKNESS OF ATTENTION OUTPUTS RESULTS FROM THE OUTPUT PROJECTION MATRIX

Among all activation types, attention outputs consistently exhibits the most rapid singular spectral decay. To investigate whether this low-rank structure originates from the attention heads outputs ($Z$), the output projection matrix ($W^O$), or their interaction, we perform a decomposition-based analysis.

Recall that the attention output is computed as $O = ZW^O$, where $Z \in \mathbb{R}^{n \times d}$ is the concatenated output of all attention heads, and $W^O \in \mathbb{R}^{d \times d}$ is a learned linear projection. To understand how the variance in $O$ (singular value spectra of O) is shaped, we analyze the variance of $O$ along an arbitrary unit direction $\hat{e} \in \mathbb{R}^d$, given by:

$$\mathrm{Var}(O\hat{e}) = \mathrm{Var}(ZW^O\hat{e}).$$

This expression highlights that the variance along $e$ is determined by two factors: the norm of $W^O\hat{e}$ and the variance of $Z$ projected onto the direction $W^O\hat{e}$. Specifically, we can rewrite the variance as:

$$\mathrm{Var}(O\hat{e}) = \mathrm{Var}(Z\hat{v}) \cdot \|v\|_2^2 \quad \text{where} \quad v = W^O\hat{e}, \ \hat{v} = \frac{v}{\|v\|_2}.$$

$$\dim\left(\bigcup_i \mathrm{span}(\mathrm{head}_i)\right) \ \leq \ \sum_i \dim\big(\mathrm{span}(\mathrm{head}_i)\big) \ = \ d_{\mathrm{head}} \cdot n_{\mathrm{head}} \ (\ = \ d_{\mathrm{model}} \text{ in standard MHSA}).$$

## 5    ACTIVE SUBSPACE INITIALIZATION FOR SPARSE AUTOENCODERS

### 5.1    EMPIRICAL CORRELATION BETWEEN LOW-RANK STRUCTURE AND DEAD FEATURES

To explore how low-rankness affects the interpretability of attention, we use the same framework and data as the original study to evaluate the LlamaScope SAEs (He et al., 2024), which provide a complete set of SAEs trained on attention output, MLP output, and residual stream [4]. We find that the number of dead features is strongly related to effective dimensions, as shown in Figure 4. This observation suggests that dead features may stem from the low-rank geometry of the activation space. We also train SAEs using different SAE hyperparameters and further systematically verify that this phenomenon is prevalent in Appendix G.

---

[4]Another prominent set of open-source SAEs, GemmaScope (Lieberum et al., 2024), train their attention SAEs on Z rather than attention output.

We refer to $\mathrm{Var}(Z\hat{v})$ as the contribution of $Z$, capturing how much variance the head output $Z$ provides in that direction, and $\|v\|_2^2$ as the contribution of $W^O$, measuring how much the output projection $W^O$ scales or suppresses that direction.

We compute and visualize the variance and these two contributions across a set of directions aligned with the right singular vectors of attention output, as shown in Figure 3. This analysis reveals that the low-rank structure of attention outputs $O$ is structurally imposed by the anisotropy of $W^O$, which compresses $Z$ into a lower-dimensional subspace. From a mechanistic perspective, an intuitive way to see this is that although each attention head contributes a $d_{\mathrm{head}}$-dimensional subspace, the superposition of heads (Jermyn et al., 2024; He et al., 2025) inherently leads to overlaps among these subspaces. We note the output of the $i^{th}$ head as $\mathrm{head}_i$. Consequently, the dimension of the MHSA output satisfies

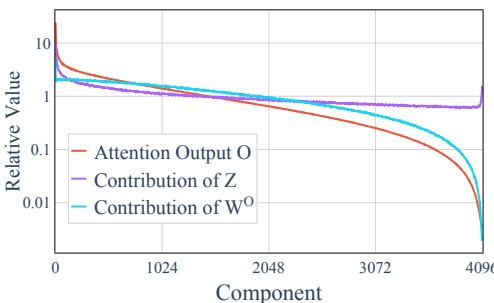

Figure 3: Decomposition of variance in attention output $O$. We analyze the contributions of the *concatenated head outputs $Z$* and the *projection matrix $W^O$* to the variance along each singular vector of $O$ ($=ZW^O$). For each component, the red value is the product of the purple and blue values. All values are normalized to a common scale and the relative value are shown. The curve of $O$ closely follow that of $Z$ for the top components, whereas its downward trend at the tail is mainly due to $W^O$ contribution.

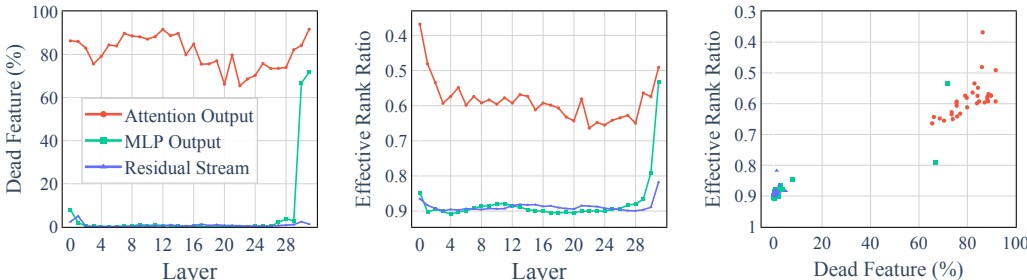

Figure 4: The number of dead features (left) and the effective rank (mid) of each activation in Llama-3.1-8B, shows a surprising consistency (right): activations with lower effective rank have more dead features, corresponding to all layers of attention output and last two layers of MLP output.

## 5.2 ACTIVE SUBSPACE INITIALIZATION FOR SPARSE AUTOENCODERS

Based on this observation, we propose *Active Subspace Initialization* (ASI), a lightweight and generalizable strategy for scaling SAEs to high capacities. Let $d$ denote the input dimension, $h$ the hidden dimension of the SAE, and $n$ the number of data points. Given activation matrices $\widetilde{A} \in \mathbb{R}^{n \times d}$ with singular value decomposition $\widetilde{A} = U\Sigma V^\top$ and $V \in \mathbb{R}^{d \times d}$ contains the right singular vectors, we select the top $d_{\mathrm{init}}$ singular vectors to define the active subspace:

$$V_{\mathrm{active}} = V_{:,:d_{\mathrm{init}}} \in \mathbb{R}^{d \times d_{\mathrm{init}}}.$$

To initialize the SAE within this subspace, we first randomly initialize the decoder weights $W_D \in \mathbb{R}^{h \times d}$ and then *project their first $d_{init}$ columns onto the active subspace*:

$$W_D \leftarrow W_D[:, : d_{\mathrm{init}}] \, V_{\mathrm{active}}^\top, \qquad W_E = W_D^\top.$$

where $W_E$ is the encoder weight matrix and $W_D$ is the decoder weight matrix. Intuitively, ASI aligns the initial SAE parameters with the active directions of the data, ensuring that SAEs start in a meaningful low-dimensional subspace. As $d_{\mathrm{init}}$ decreases from the full space dimension[5] within

---

[5]Setting $d_{init}$ equal to the full space dimension is equivalent to not using Active Subspace Initialization.

a certain range, the number of dead features in the Attention Output SAE rapidly drops, with a corresponding improvement in MSE and Delta LM loss[6] (Figure 5). We refer readers to Appendix F for complete SAE training details. Further ablation studies and the pseudocode of ASI are contained in Appendix K and Appendix L, respectively.

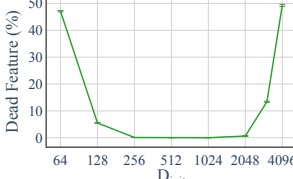 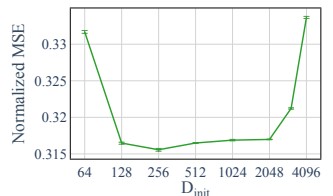 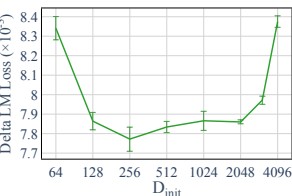

Figure 5: After using ASI, proportion of dead features (left), normalized MSE (mid) and Delta LM loss (right) across different $d_{init}$ for activations with a full space dimension of 4096. All experiments repeat 3 times using different random seeds and error bars indicate mean ± std.

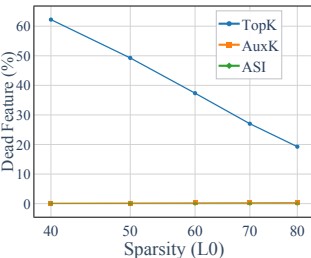 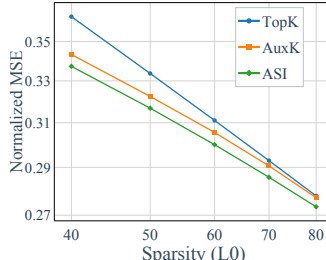 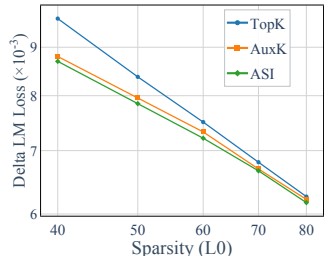

Figure 6: At a fixed number of features ($n = 32768$), ASI (TopK SAE with Active Subspace Init) achieves a better reconstruction-sparsity trade-off than TopK (standard TopK SAE) and AuxK (TopK SAE with auxiliary loss). A similar trend is observed in its impact on Delta LM Loss. All experiments repeat 3 times using different random seeds and show the mean. The results of std are in Appendix H due to resolution constraints.

Using **Active Subspace Initialization** offers several benefits:

**Reduced dead features and Enhanced Sparsity-Reconstruction Frontier Without Additional Compute**    It achieves near-zero dead features and slightly superior results compared to the auxiliary loss approach (AuxK), at no additional computational cost of the same order. (Figure 6).

**Optimal Scaling Characteristics**    Our approach demonstrates optimal scaling behavior across various SAE training methods. It outperforms TopK and AuxK in any evaluated scale, from 16K to 1M features (Section 5.3).

**General Applicability**    The technique maintains applicability to diverse architectural variants and activation functions, as it operates directly on the intrinsic properties of activations. This generalizability is further explored in Section 5.4 and Appendix M.

We conduct a statistical significance test in Appendix I to demonstrate the statistical significance of our conclusions. We validate the effectiveness of ASI across different layers, models, and datasets in Appendix J. We further compare the features of TopK and ASI in Appendix O

## 5.3    SCALING LAWS

To understand how our method scales, we evaluate performance as the number of SAE features increases from 16K to 1M, keeping other hyperparameters fixed (details in Appendix F).

---

[6]This metric is defined as the difference between the original language model loss and the loss when the SAE is inserted at the corresponding position, evaluated over 1 million tokens.

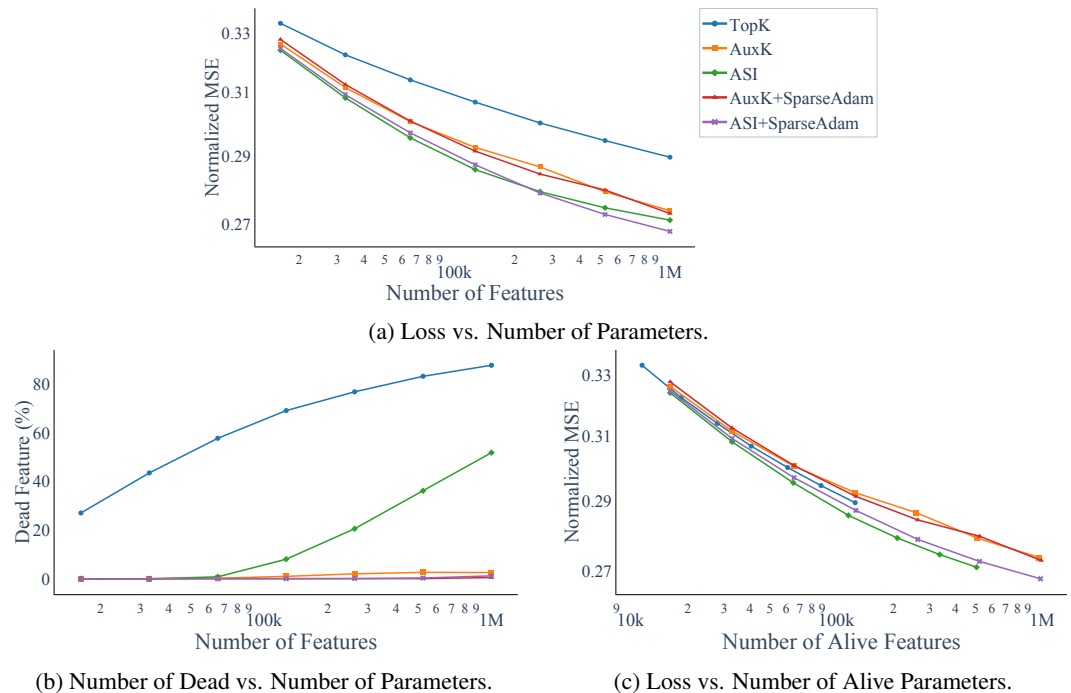

(a) Loss vs. Number of Parameters.

(b) Number of Dead vs. Number of Parameters.

(c) Loss vs. Number of Alive Parameters.

Figure 7: Scaling results of TopK SAEs and their variants enhanced with *AuxK*, *Active Subspace Init*, and *SparseAdam*–all trained on attention output from the middle layer of Llama-3.1-8B. (A) Loss at convergence across different feature counts: Active Subspace Init consistently achieves lower reconstruction error than TopK and AuxK. Active Subspace Init with SparseAdam achieves the best at large scale. (B) Dead features: Active Subspace Init reduces dead features compared to TopK, but still retains many at extremely large scales. Enhanced with SparseAdam, dead features can be reduced to less than 1%. (C) Loss across different number of alive features: Active Subspace Init achieves the most efficient utilization of alive features, while AuxK shows the lowest efficiency. Details in Section 5.3.

**Active Subspace Init improves reconstruction.** As shown in Figure 7a, Active Subspace Init consistently outperforms TopK and AuxK across all scales.

**Caveat: some dead features remain at extremely large scales in Active Subspace Init.** Figure 7b shows that, when scaling to extremely large feature counts, Active Subspace Init produces more dead features than AuxK. However, reconstruction performance remains better, indicating that the revived features from AuxK contribute little to actual reconstruction quality(Figure 7c).

**Use Active Subspace Init with SparseAdam further improves performance.** Prior work (Bricken et al., 2023a) identified *stale momentum* as a key factor in dead feature formation. Building on this insight, we propose using **SparseAdam**, an optimizer specifically designed for sparse activation settings. By updating only the moments and parameters corresponding to non-zero gradients, SparseAdam naturally avoids stale momentum and thus mitigates the dead feature issue. As shown in Figures 7a, 7b, combining Active Subspace Init with SparseAdam substantially reduces dead features while reaching the lowest reconstruction error. While orthogonal to our initialization method, this choice provides a practical complement that further stabilizes training when scaling SAEs to very large capacities. We discuss more about *stale momentum* and **SparseAdam** in Appendix N.

### 5.4 GENERALIZE TO SPARSE REPLACEMENT MODELS

Recent work by He et al. (2025) reports that Lorsa, a sparse replacement model for attention layers, exhibits a high proportion of dead parameters. We hypothesize that the low-rank structure of attention outputs contributes significantly to this phenomenon.

To evaluate this, we apply **Active Subspace Initialization** to Lorsa. This modification reduces the proportion of dead parameters significantly under identical hyperparameter settings (Figure 8), while also improving the reconstruction. Specifically, we initialize the decoder (corresponding to $W^O$ in Lorsa) within the active subspace of each original MHSA head, while the encoder (corresponding to $W^V$ in Lorsa) is aligned with the corresponding input-side directions of the decoder. The pseudocode of ASI used on Lorsa are provided in Appendix L .

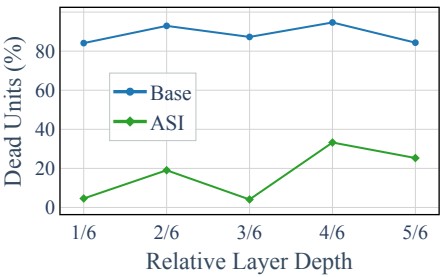 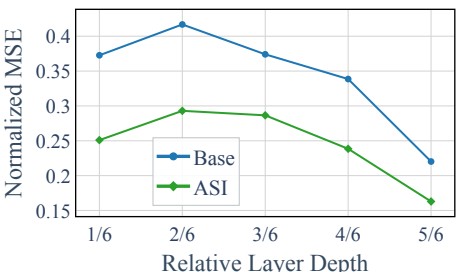

Figure 8: Effect of Active Subspace Initialization on reducing dead parameters in attention replacement model.

## 6    DISCUSSION

**When to Use Active Subspace Initialization**    For activation sites that do not exhibit clear low-rank structures (e.g., residual stream), our method yields only limited gains (Appendix K.2). The singular value spectrum provides a more reliable indicator: when the spectrum decays rapidly in its tail, applying ASI is likely to be effective.

**Causality between Low-Rank Structure and Dead Features**    We observed a strong correlation between low-rank structure and the emergence of dead features (Section 5), but the causal mechanism remains unclear. It may be tied to optimization dynamics or feature competition, yet a precise explanation is lacking. A deeper understanding of this relationship could also explain why ASI is less effective under high-$\ell_0$ warm-up regimes (Appendix M). We leave this important line of inquiry to future work.

## 7    CONCLUSION

We identified the low-rank structure of attention outputs as a fundamental property of Transformer models and a key cause of dead features in sparse dictionary learning. Our proposed *Active Subspace Initialization* method addresses this by aligning SAE features with the intrinsic geometry of activations, reducing dead features while improving reconstruction quality. The approach generalizes beyond SAEs to sparse replacement models.

## ETHICS STATEMENT

This work relies solely on computer-based experiments with publicly available open-source language models. No human subjects, private data, or sensitive information were involved. We followed the ICLR Code of Ethics and do not foresee any ethical concerns related to our study, as the research does not pose risks of discrimination, privacy violations, or other harmful applications.

## REPRODUCIBILITY STATEMENT

We have made extensive efforts to ensure the reproducibility of our results. All model configurations, training settings, and evaluation protocols are described in detail in the main text and Appendix B, F, L. Furthermore, we provide an anonymous link `https://anonymous.4open.science/r/Language-Model-SAEs-2B1D/` to the full source code and scripts used in our experiments, enabling others to reproduce our findings.

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

## A  DECLARATION OF LLM USAGE

We acknowledge the use of large language models (LLMs) to assist in the preparation of this manuscript. Specifically, **ChatGPT** and **Claude** were utilized during the writing and coding process for the exclusive purpose of improving text clarity, grammar, overall fluency, and for generating boilerplate code structures. All ideas, theoretical developments, experimental designs, results, analyses, and scientific conclusions are entirely our own. The LLMs acted solely as assistive tools and were not involved in any aspect of the intellectual or scientific work.

## B  ACTIVATION SOURCES

The spectral characteristics of activations vary substantially across model architectures, datasets, and positional contexts. Below, we describe the experimental configurations used to support a broad and representative analysis.

**Models**  We study three large language models of different families–Llama-3.1-8B[7], Qwen3-8B[8], and Gemma-2-9B[9]–all based on the Transformer architecture. This allows us to assess the robustness of spectral properties under varying model training configurations.

**Datasets**  To investigate how dataset diversity affects activation spectra, we select three datasets with varying linguistic and domain characteristics: (1) SlimPajama[10], an English corpus comprising web text, Github, Arxiv and other sources and (2) CCI3-Data[11], a Chinese dataset with broad domain coverage.

---

[7]https://huggingface.co/meta-llama/Llama-3.1-8B
[8]https://huggingface.co/Qwen/Qwen3-8B
[9]https://huggingface.co/google/gemma-2-9b
[10]https://huggingface.co/datasets/cerebras/SlimPajama-627B
[11]https://huggingface.co/datasets/BAAI/CCI3-Data

**Activation Positions** We analyze three types of activations: (1) attention output, (2) MLP output, and (3) residual stream (post layer).

## C FORMAL DEFINITION OF *Fraction of Loss Recovered*

Given the original language model cross-entropy loss is $loss_{\text{original}}$, the loss after ablating the activation at a specific position to zero is $loss_{\text{zero}}$, and the loss after replacing the original activation projected to the subspace spanned by first $n$ singular vectors is $loss_{\text{recovered}}$. Then, for these $n$ components, the fraction loss recovered is calculated as:

$$\frac{loss_{\text{zero}} - loss_{\text{recovered}}}{loss_{\text{zero}} - loss_{\text{original}}}$$

## D ERROR ANALYSIS IN SINGULAR VALUE DECOMPOSITION

For the attention output of layer 15 of Llama-3.1-8B, we performed 5 times of singular value decompositions, using different 10 million tokens for each, and calculated the Coefficient of Variation (CV) for each singular value across these 5 runs. The maximum CV was only $4.9 \times 10^{-3}$, and the mean and standard deviation of the effective rank computed from these 5 SVD results were 2523.165 and 0.404, respectively, with a CV of $1.5 \times 10^{-4}$. These error experiments show that using 10 million tokens for singular value decomposition is sufficiently stable.

## E MORE SINGULAR SPECTRUM AND EFFECTIVE RANK RESULTS

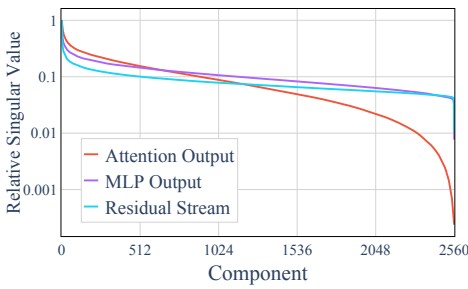

(a) Middle layer of pythia-2.8b; SlimPajama
Effective Dimensionality: Attention Output 1670;
Mlp Output 2252; Residual Stream 2327

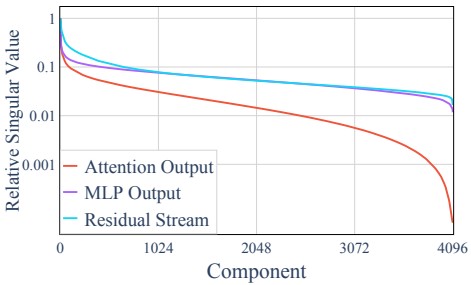

(b) Middle layer of Qwen3-8B; CCI3-Data
Effective Dimensionality: Attention Output 2356;
Mlp Output 3558; Residual Stream 3140

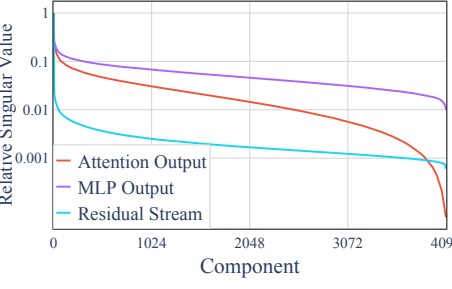

(c) Middle layer of Qwen3-8B; SlimPajamaGithub
Effective Dimensionality: Attention Output 2410;
Mlp Output 3495; Residual Stream 2000

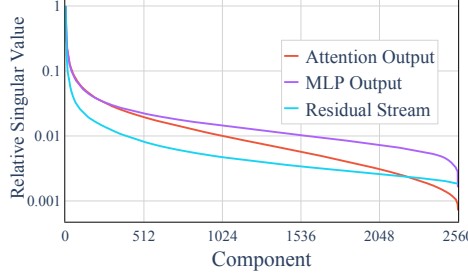

(d) Middle layer of Qwen3-4B; SlimPajama
Effective Dimensionality: Attention Output 1122;
Mlp Output 1485; Residual Stream 990

Figure 9

## E.1 ACROSS MODELS AND DATASETS

We present relative singular values for some other model-dataset pairs in Figure 9. Models include pythia-2.8b[12]. Datasets include SlimPajamaGithub (subset of SlimPajama) and CCI3-Data.

## E.2 ACROSS ACTIVATION POSITION

We present the effective rank for activations that are commonly used to train SAEs in Figure 10, including **Concatenated Outputs of all Attention Heads** (Z), attention output, the hidden activations of MLP (post activation fuction), MLP output, residual stream. All effective ranks are computed on SlimPajama.

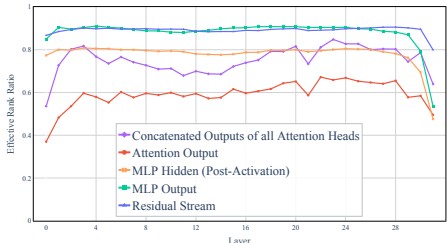

(a) The number of the effective rank of each activation in Llama-3.1-8B

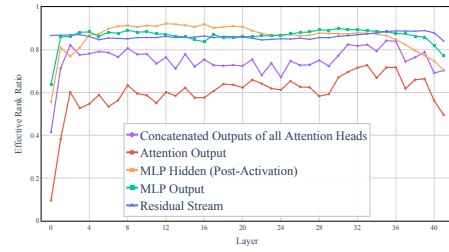

(b) The number of the effective rank of each activation in Gemma-2-9B

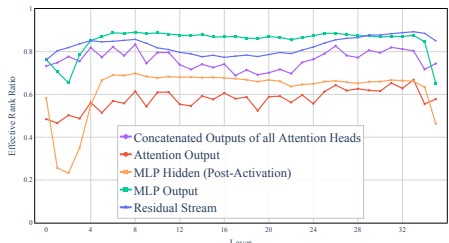

(c) The number of the effective rank of each activation in Qwen3-8B

Figure 10

## F SAE TRAINING DETAILS

We train SAEs as the following description.

### F.1 HYPERPARAMETERS

**Model, Dataset, Layer, Pos**   Llama-3.1-8B, SlimPajama, 15(index start at 0), attention output.

**Sparsity**   We empirically set $k = 50$ for a reasonable sparsity following He et al. (2024), except the experiments for sweeping $k$.

**Dictionary Size**   We empirically set $n_{features} = 32768$ which is $8 \times d_{model}$, except the experiments for sweeping dictionary size (scaling law).

**Batch Size**   We empirically set the batch size to 4096.

---

[12]https://huggingface.co/EleutherAI/pythia-2.8b

**Optimizer**    We use the Adam and SparseAdam optimizer, both with $\beta_1 = 0.9$, $\beta_2 = 0.999$, and $\epsilon = 10^{-8}$. Unless otherwise specified, Adam is used by default.

**Learning Rate**    We search best learning rate again because of the change of batch size. The learning rate for **Adam** and **SparseAdam** is sweeped separately in [1e$-$5, 2e$-$5, 4e$-$5, 6e$-$5, 8e$-$5, 1e$-$4, 2e$-$4, 4e$-$4], and we ultimately use 4e$-$5 for **Adam** and 6e$-$5 for **SparseAdam**. We employ a three-phase learning rate schedule consisting of a linear warm-up, a stable phase, and a linear decay. The learning rate increases linearly from zero to its maximum value over the first 500 steps, remains constant during the intermediate phase, and then decays linearly to 1% of the maximum value over the final 20% of the total training steps.

**AuxK**    We follow Gao et al. (2024) to set auxiliary loss coefficient $\alpha$ as $\frac{1}{32}$. We sweep the $k_{aux}$ in [256, 512, 1024, 2048] and finally choose 512. We also sweep $\alpha$ and find the results are less sensitive to $\alpha$ in a reasonable interval.

**Dimension of Subspace for SAE Initialization ($d_{init}$)**    We use 768 for all experiments, except the experiments for sweeping $d_{init}$. We refer readers to Appendix K.3 for the reason.

**Total Tokens**    We use 800M tokens for each SAE training.

## F.2    COLLECTING ACTIVATIONS

We truncate each document to 1024 tokens and prepend a <bos> token to the beginning of each document. During training, we exclude the activations corresponding to the <bos> and <eos> tokens.

It has been observed that activations from different sequence positions within the same document are often highly correlated and may lack diversity. To mitigate this issue, it is common to introduce randomness into the training data. Our shuffling strategy maintains a buffer that is reshuffled whenever the buffer is refilled.

## F.3    INITIALIZATION

The decoder columns $W^{dec}_{:,i}$ are initialized uniformly, and the optimal norm for them is found through a grid search to minimize the initial reconstruction loss. We find that the specific initialization norm has little impact, as long as in a reasonable scope. For example, initializing $W^{dec}_{:,i}$ uniformly with a fixed bound, as in Conerly et al. (2025), yields similar results. The encoder weights $W^{enc}$ are initialized as the transpose of $W^{dec}$, while both the encoder bias $b^{enc}$ and decoder bias $b^{dec}$ are set to zero.

## F.4    JUMPRELU SAEs

We trained JumpReLU SAEs (Rajamanoharan et al., 2024b) under two distinct hyperparameter configurations: one maintaining a consistently low $\ell_0$ value throughout training, and another where $\ell_0$ is gradually decreased from a higher initial value. Unless otherwise specified, all JumpReLU SAEs were trained using the same settings as Conerly et al. (2025), which corresponds to the latter configuration. The key modifications for the former setting are as follows: (1) we initialized the encoder bias to zero instead of applying the heuristic that equalizes feature activation counts at initialization, and (2) we kept the sparsity coefficient fixed rather than employing a global warm-up schedule. As a result, the $\ell_0$ sparsity level started at a relatively low value early in training. This design is critical to our approach: we observed that if the model remains in a high-$\ell_0$ regime (e.g., on the order of $d_{\text{model}}/2$) for an extended period before sparsity increases, the feature directions tend to drift away from the active subspace during this phase, thereby diminishing the effectiveness of our method (Appendix M.1).

## G  ADDITIONAL ANALYSIS ON EFFECTIVE DIMENSIONALITY AND DEAD FEATURES

This section provides additional experimental evidence supporting the claim that *low intrinsic dimensionality strongly correlates with a higher proportion of dead features* in sparse autoencoders (SAEs). In Figure 4, we report the effective rank of the residual stream, attention output, and MLP output at each layer of `Llama-3.1-8B`, along with the proportion of dead features in the SAEs trained on these activations. All SAEs were obtained from `Llamascope` (He et al., 2024), which uses the same dictionary size (32768 features) and sparsity level ($L_0 = 50$).

To provide systematic analysis, we additionally train SAEs across multiple dictionary sizes (16384, 32768, 65536) and sparsity levels ($L_0 \in \{32, 64, 128\}$) following Appendix F The SAEs are trained on the activations of `Llama-3.1-8B`, and the corresponding effective ranks of these activations can be found in Figure 4 of the main paper. Tables 1–3 summarize the proportion of dead features across these configurations.

Across all settings, attention outputs consistently show substantially higher dead-feature ratios than residual streams. This trend holds even when the dictionary size varies by a factor of four and the sparsity level varies by a factor of four.

Table 1: Proportion of dead features for $L_0 = 32$ across different dictionary sizes.

| Activation (Effective Rank) | 16384 | 32768 | 65536 |
|---|---|---|---|
| Layer 7 attention (2351) | 84.80% | 90.31% | 94.02% |
| Layer 7 residual (3664) | 1.61% | 6.69% | 17.10% |
| Layer 15 attention (2506) | 68.70% | 79.86% | 87.22% |
| Layer 15 residual (3611) | 27.01% | 45.70% | 61.33% |
| Layer 23 attention (2654) | 58.45% | 70.48% | 78.81% |
| Layer 23 residual (3634) | 0.13% | 0.26% | 1.35% |

Table 2: Proportion of dead features for $L_0 = 64$ across different dictionary sizes.

| Activation (Effective Rank) | 16384 | 32768 | 65536 |
|---|---|---|---|
| Layer 7 attention (2351) | 66.97% | 75.65% | 82.96% |
| Layer 7 residual (3664) | 0.02% | 0.06% | 0.15% |
| Layer 15 attention (2506) | 41.65% | 54.68% | 67.43% |
| Layer 15 residual (3611) | 1.73% | 7.96% | 18.83% |
| Layer 23 attention (2654) | 41.58% | 56.70% | 67.05% |
| Layer 23 residual (3634) | 0.15% | 0.09% | 0.08% |

Table 3: Proportion of dead features for $L_0 = 128$ across different dictionary sizes.

| Activation (Effective Rank) | 16384 | 32768 | 65536 |
|---|---|---|---|
| Layer 7 attention (2351) | 49.85% | 56.96% | 64.83% |
| Layer 7 residual (3664) | 0.00% | 0.00% | 0.02% |
| Layer 15 attention (2506) | 15.09% | 25.61% | 37.11% |
| Layer 15 residual (3611) | 0.07% | 0.12% | 0.44% |
| Layer 23 attention (2654) | 21.90% | 39.02% | 52.68% |
| Layer 23 residual (3634) | 0.14% | 0.09% | 0.07% |

## H  COMPLETE RESULTS OF SAE METRICS

We use 3 different random seeds for all experiments in Figure 6 and compute the mean values and the standard deviations of each metrics, as shown in Table 4.

Table 4: Comparison of Base, Auxk, and ASI across different $L_0$ settings. Numbers show mean $\pm$ std over 3 seeds.

| $L_0$ | Metric | Base | Auxk | ASI |
|---|---|---|---|---|
| 40 | Dead Feature | $20395.00 \pm 72.77$ | $37.00 \pm 5.20$ | $10.67 \pm 1.15$ |
|  | Normalized MSE | $0.36323 \pm 0.00018$ | $0.34328 \pm 0.00011$ | $0.33724 \pm 0.00022$ |
|  | Delta LM Loss ($\times 10^{-3}$) | $9.650 \pm 0.040$ | $8.801 \pm 0.053$ | $8.697 \pm 0.033$ |
| 50 | Dead Feature | $16144.33 \pm 129.52$ | $54.67 \pm 5.51$ | $4.00 \pm 1.73$ |
|  | Normalized MSE | $0.33367 \pm 0.00014$ | $0.32241 \pm 0.00020$ | $0.31680 \pm 0.00006$ |
|  | Delta LM Loss ($\times 10^{-3}$) | $8.375 \pm 0.029$ | $7.958 \pm 0.032$ | $7.847 \pm 0.050$ |
| 60 | Dead Feature | $12239.33 \pm 165.51$ | $76.00 \pm 9.17$ | $2.67 \pm 1.53$ |
|  | Normalized MSE | $0.31106 \pm 0.00007$ | $0.30555 \pm 0.00007$ | $0.30000 \pm 0.00005$ |
|  | Delta LM Loss ($\times 10^{-3}$) | $7.503 \pm 0.059$ | $7.325 \pm 0.017$ | $7.214 \pm 0.026$ |
| 70 | Dead Feature | $8854.67 \pm 40.51$ | $115.33 \pm 15.37$ | $1.67 \pm 0.58$ |
|  | Normalized MSE | $0.29295 \pm 0.00008$ | $0.29064 \pm 0.00008$ | $0.28575 \pm 0.00013$ |
|  | Delta LM Loss ($\times 10^{-3}$) | $6.805 \pm 0.074$ | $6.694 \pm 0.021$ | $6.664 \pm 0.066$ |
| 80 | Dead Feature | $6311.33 \pm 55.77$ | $109.67 \pm 11.50$ | $1.00 \pm 1.73$ |
|  | Normalized MSE | $0.27787 \pm 0.00003$ | $0.27715 \pm 0.00003$ | $0.27341 \pm 0.00003$ |
|  | Delta LM Loss ($\times 10^{-3}$) | $6.259 \pm 0.005$ | $6.217 \pm 0.028$ | $6.168 \pm 0.033$ |

# I STATISTICAL SIGNIFICANCE TEST

To assess whether the performance improvements introduced by ASI are statistically significant, we conducted a comprehensive significance analysis across multiple evaluation metrics.

## I.1 EXPERIMENTAL SETUP

We evaluate the statistical significance of performance differences between ASI and two baseline methods (TopK and AuxK) under the following controlled setting:

- **Model / Layer / $L_0$ / Dictionary Size:** Llama-3.1-8B, Layer 15, $L_0 = 50$, dictionary size = 32,768.
- **Number of runs:** 15 independent trials for each method, each with a different random seed.
- **Evaluation metrics:** Dead Feature Count, Normalized MSE, and $\Delta$ LM Loss. All metrics follow a "lower is better" criterion.
- **Comparisons performed:** ASI vs. TopK and ASI vs. AuxK for all three metrics.

## I.2 HYPOTHESIS TESTING FRAMEWORK

For each metric and each baseline method, we perform Welch's t-test (also known as Welch's unequal-variance t-test), which does not assume equal variances between groups. For ASI and a given baseline method, we test the following hypotheses:

$$H_0 : \mu_{\text{ASI}} \geq \mu_{\text{baseline}} \quad \text{(ASI is worse than or equal to the baseline),}$$
$$H_1 : \mu_{\text{ASI}} < \mu_{\text{baseline}} \quad \text{(ASI outperforms the baseline).}$$

This is a one-tailed test, as we explicitly test whether ASI achieves significantly lower metric values.

We use the following SciPy function for all tests:

```
scipy.stats.ttest_ind(ASI, baseline, equal_var=False, alternative='less').
```

## I.3 RESULTS

Table 5 reports the resulting p-values for all comparisons. A smaller p-value indicates stronger evidence that ASI outperforms the baseline.

Table 5: Welch's t-test results for ASI compared with TopK and AuxK across 15 random seeds.

| Comparison | Metric | p-value |
|---|---|---|
| ASI vs. TopK | Dead Feature Count | $3.26 \times 10^{-35}$ |
| | Normalized MSE | $2.99 \times 10^{-40}$ |
| | $\triangle$ LM Loss | $1.14 \times 10^{-23}$ |
| ASI vs. AuxK | Dead Feature Count | $4.33 \times 10^{-15}$ |
| | Normalized MSE | $6.33 \times 10^{-40}$ |
| | $\triangle$ LM Loss | $1.27 \times 10^{-6}$ |

Across all evaluation metrics and both baseline methods, the p-values are far below standard significance thresholds (e.g., $\alpha = 0.05$). Therefore, we reject the null hypothesis for all comparisons. These results demonstrate that the improvements achieved by ASI are statistically significant and robust across random seeds.

## J ADDITIONAL EVALUATION ACROSS LAYERS, MODELS, AND DATASETS

To assess the robustness and generality of ASI, we extend our experiments beyond the primary configuration used in the main paper (Llama-3.1-8B, Layer 15, SlimPajama). In particular, we investigate whether the advantages of ASI over baseline approaches (TopK and AuxK) persist across different layers, models, and datasets. We consider this evaluation essential, as mechanisms in sparse autoencoding can vary substantially across architectural depth, data distribution, and model family.

### J.1 EVALUATION ON LLAMA-3.1-8B ACROSS MULTIPLE LAYERS

We first evaluate ASI on two additional layers of Llama-3.1-8B (Layers 7 and 23), using the SlimPajama dataset. Activations are taken from the attention output. Results are summarized in Table 6.

Table 6: Performance of ASI and baseline methods on Llama-3.1-8B Layers 7 and 23. Lower values indicate better performance.

| Layer | Dead Feature Count | | | Normalized MSE | | | $\triangle$ LM Loss ($\times 10^{-3}$) | | |
|---|---|---|---|---|---|---|---|---|---|
| | Base | AuxK | ASI | Base | AuxK | ASI | Base | AuxK | ASI |
| 7 | 25836 | **98** | 5308 | 0.32870 | 0.29800 | **0.28882** | 5.414 | 4.589 | **4.430** |
| 23 | 15542 | 332 | **27** | 0.21302 | 0.20235 | **0.20141** | 1.942 | 1.865 | **1.845** |

We observe that ASI consistently achieves the lowest reconstruction error (Normalized MSE) and the smallest degradation in language modeling performance ($\triangle$ LM loss). For Layer 7, ASI still retains a number of dead features, which may be attributed to its smaller effective rank (2351) compared to Layer 23 (2654). Since we use a fixed $d_{\text{init}}$ across layers, this mismatch can lead to remaining dead features. Despite this, ASI still achieves the lowest MSE on both layers.

### J.2 EVALUATION ON QWEN3-8B ACROSS LAYERS AND A NEW DATASET

To further test cross-model and cross-dataset generality, we conduct experiments on the Qwen3-8B model using the fineweb-edu dataset, again using attention-output activations. Results for Layers 8 and 26 are shown in Table 7.

The results again confirm that ASI achieves the lowest reconstruction error and the smallest increase in LM loss across both layers. The near-dead-feature-free representation produced by AuxK is also observed, but ASI consistently outperforms AuxK in reconstruction quality and LM preservation.

Table 7: Performance comparison on Qwen3-8B Layers 8 and 26 using the fineweb-edu dataset. Lower values indicate better performance.

| Layer | Dead Feature Count | | | Normalized MSE | | | Δ LM Loss ($\times 10^{-3}$) | | |
|---|---|---|---|---|---|---|---|---|---|
| | Base | AuxK | ASI | Base | AuxK | ASI | Base | AuxK | ASI |
| 8 | 19048 | **4** | 7 | 0.31228 | 0.28849 | **0.28533** | 2.9561 | 2.5911 | **2.5667** |
| 26 | 16286 | **66** | 566 | 0.30090 | 0.28127 | **0.28087** | 1.3406 | 1.2980 | **1.2922** |

## J.3 SUMMARY

Across all tested configurations—spanning multiple layers, two large language model families, and two datasets—ASI exhibits consistent advantages over baseline methods. These evaluations provide strong empirical evidence that the benefits of ASI are not confined to a specific layer, model, or dataset, but instead generalize across diverse settings.

## K ABLATION STUDY

### K.1 ACTIVE SUBSPACE INIT VS RANDOM SUBSPACE INIT

We employ random subspace initialization as a baseline and observe that it consistently degrades SAE training across all metrics, as shown in Figure 11.

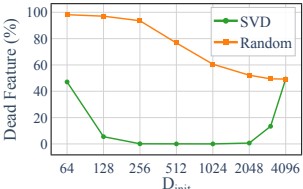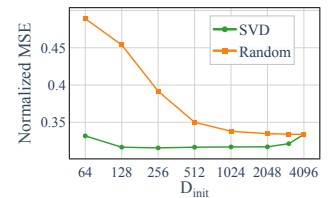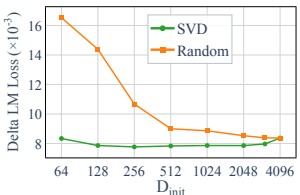

Figure 11: For activations with a full space dimension of 4096, proportion of dead features (left), normalized MSE (mid) and Delta LM loss (right) across different subspace dimensions. Random subspaces are used as the baseline, whereas only initialization with the active subspace yields improvement.

### K.2 APPLY ON NEAR-FULL-RANK ACTIVATION

We also apply Active Subspace Initialization (ASI) to near-full-rank activations, such as those in the residual stream, to evaluate its generality. When training an SAE on the post-layer-15 residual stream of Llama-3.1-8B, we find ASI yields minimal gains (Figure 12). This is consistent with our expectation, as these activations inherently exhibit a lower rate of dead features even with standard initialization.

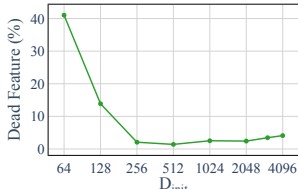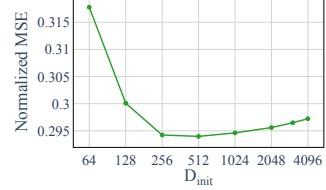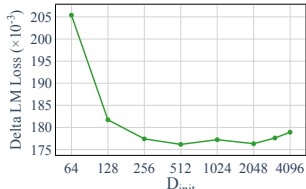

Figure 12: For activations with a full space dimension of 4096, proportion of dead features (left), normalized MSE (mid) and Delta LM loss (right) across different subspace dimensions. Random subspaces are used as the baseline, whereas only initialization with the active subspace yields improvement.

### K.3 CHOICE OF THE INITIAL DICTIONARY SIZE $d_{init}$

As shown in Figure 5, $d_{init}$ is a hyperparameter with a wide range of acceptable values (from 256 to 2048). We hypothesize that the performance degradation at very low values occurs because an excessively small dictionary fails to span the subspace containing critical information needed for effective reconstruction.

## L PSEUDO-CODE FOR IMPLEMENTING ACTIVE SUBSPACE INIT

Below is a PyTorch-style pseudo-code for Active Subspace Initialization.

**Use on SAE**

```
# X: activation batch [batch_size, d_model]
# W_E: decoder weight [d_model, d_sae]
# W_D: decoder weight [d_sae, d_model], initialized uniformly
# d_active_subspace: target subspace dimension

# 1. Demean the activations
demeaned_X = X - X.mean(dim=0) # [batch_size, d_model]

# 2. Compute SVD
U, S, V = torch.svd(demeaned_label) # V: [d_model, d_model]

# 3. Take top-d_init singular vectors
proj_weight = V[:, :d_init]  # [d_model, d_init]

# 4. Fold projection into decoder weights
W_D.copy_(W_D[:, :d_init] @ proj_weight.T)

# 5. Init W_E with W_D.T
W_E.copy_(W_D.T)
```

**Use on Lorsa**

```
# X: activation batch [batch_size, seq_len, d_model]
# mhsa: pretrained MHSA module with W_O and W_V
# W_O, W_V: Lorsa decoder/encoder weights
# d_head: dimension of original MHSA heads
# n_heads: number of original MHSA heads
# n_lorsa_heads: number of Lorsa heads

# 1. Run MHSA to capture per-head outputs
Z = mhsa.compute_z(X) # [batch_size, seq_len, n_heads, d_head]
output_per_head = torch.einsum('bsnh, nhd->bsnd', Z, mhsa.W_O)

# 2. For each original head
# initialized some lorsa heads into it's active subspace
rate = n_lorsa_heads // n_heads
for mhsa_index in range(n_heads):
    head_slice = [rate*mhsa_index:rate*(mhsa_index+1)]
    # [B, S, d_model]
    output=output_per_head[:, :, mhsa_index, :]
    output_flat=output.flatten(0, 1) # [B*S,d_model]

    # 3.1 Demean
    demeaned_output=output_flat - output_flat.mean(dim=0)
```

```
# 3.2 Compute SVD
U, S, V = torch.svd(demeaned_output)

# 3.3 Take top-d_head singular vectors
proj_weight = V[:, :d_head]

# 3.4 Update part of decoder weights W_O
W_O[head_slice] = W_O[head_slice, :d_head] @ proj_weight.T

# 3.5 Update part of encoder weights W_V
head_trans = mhsa.W_V[mhsa_index] @ mhsa.W_O[mhsa_index]
W_V[head_slice] = W_O[head_slice] @ head_trans.T
```

The strategy of initialize $W_V$ in Lorsa is a method like the **tied initialization** used in SAEs to ensure alignment between feature encoding and decoding[13]. This approach has been shown to be crucial for reducing dead features in SAEs (Gao et al., 2024). We think the same thought could also be used to improve the replacement model for MLP (trancoder and cross layer transcoder), which we leave a deeper investigation to future work.

## M   USE ASI ON OTHER ACTIVATION FUNCTIONS

### M.1   JUMPRELU

Another wildly used activation fuction is Jumprelu (Rajamanoharan et al., 2024b). We trained the Jumprelu SAEs under two different hyperparameter settings: one with a consistently low $\ell_0$ value and another where $\ell_0$ gradually decreases from a higher initial value, as described in Appendix F.4. We observed that our method is effective in the former case (Figure 13) but shows little improvement in the latter (Figure 14). We train these SAEs following Appendix F.

For cases where one follows a schedule that gradually reduces $\ell_0$ from a high initial value, we recommend first applying PCA to reduce the dimensionality of the data. The SAE can then be trained on the reduced representation until the $\ell_0$ level reaches the target range. Afterwards, the PCA projection matrix can be folded into the model parameters, and training can continue in the original space. This achieves a similar effect without the drawbacks of prolonged training in the high-$\ell_0$ regime.

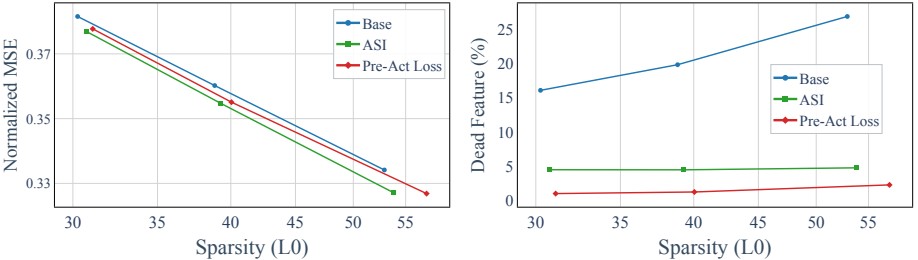

Figure 13: For the attention output of the middler layer of Llama-3.1-8B, using ASI on JumpRelu SAEs which has a low initial $\ell_0$ is effective. Details in Appendix M.1

### M.2   TOPK WITH K ANNEAL

To enhance the finding in Section M.1, we conduct experiments on a variant of TopK, which sets K to a high value and then lets it decrease during training (He et al., 2024). We find ASI also fails in this case (Figure 15).

---

[13]"Match" means encoder can be initialized to predict relatively accurate feature activation values for decoder.

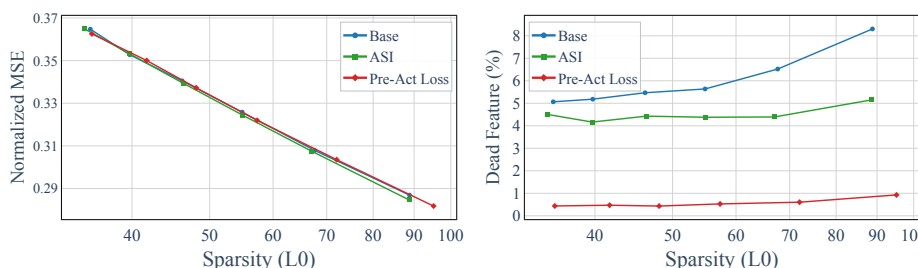

Figure 14: For the attention output of the middler layer of Llama-3.1-8B, using ASI on JumpRelu SAEs which has a high initial $\ell_0$ than gradually decreasing shows little improvement. Details in Appendix M.1

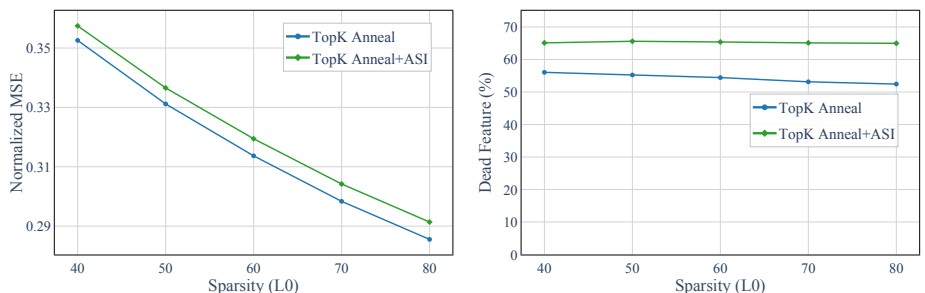

Figure 15: For the attention output of the middler layer of Llama-3.1-8B, using ASI on TopK SAEs which sets K to a high value and then lets it decrease during training fails. Details in Appendix M.2

## N  STALE MOMENTUM AS ANOTHER ROOT CAUSE OF DEAD FEATURES

Recent work by Bricken et al. (2023a) identifies *stale momentum* as a key cause to dead feature formation. Specifically, when a feature remains inactive over training steps, its associated optimizer momentum continues to accumulate. If the feature activates, the stale momentum results in disproportionately large updates, destabilizing training and potentially suppressing that feature permanently.

To directly address this, we adopt *SparseAdam*, an optimizer tailored for sparse activation settings, designed for more efficient use of compute and memory. SparseAdam updates both parameters and moments only when the corresponding feature is active. This could effectively prevent the harmful accumulation of stale momentum. Empirically, we observe that this change substantially reduces the rate of dead feature formation in large-scale SAE training. We believe that this is a core technique for scaling sparse dictionary methods, as stale momentum is a common problem for them.

## O  COMPARE FEATURES OF SAES WITH AND WITHOUT ASI

### O.1  MONOSEMANTICITY

We conducted an additional analysis to assess the degree of monosemanticity exhibited by features learned by the base TopK SAE and the ASI-enhanced SAE. Following the rubric of Hoagy Cunningham (2024), we performed a blinded evaluation of 100 randomly sampled features and recorded the semantic consistency scores assigned to each feature.

For clarity, we reproduce below the scoring rubric used for evaluating activation consistency:

- **5**: Clear pattern with no deviating examples
- **4**: Clear pattern with one or two deviating examples
- **3**: Clear overall pattern but quite a few examples not fitting that pattern

- **2**: Broad consistent theme but lacking structure
- **1**: No discernible pattern

To provide transparency, Tables 8 and 9 summarize the distribution of scores for TopK and ASI.

Across both models, we found no statistically significant differences in score distributions at this scale of analysis. Variations between the two SAE variants were marginal and did not indicate systematic differences in feature quality. This result is consistent with our expectations, as our method does not modify the SAE architecture and is not designed to intervene in how features are formed. Consequently, the potential risk of degrading feature quality remains low.

Table 8: Score distribution for features from the ASI-enhanced SAE.

| Score | Count |
|-------|-------|
| 5 | 17 |
| 4 | 14 |
| 3 | 6 |
| 1 | 8 |
| 2 | 5 |

Table 9: Score distribution for features from the base TopK SAE.

| Score | Count |
|-------|-------|
| 5 | 18 |
| 4 | 12 |
| 3 | 6 |
| 2 | 6 |
| 1 | 8 |

## O.2 ANALYSIS OF SAE FEATURES IN THE DEAD SUBSPACE

To assess whether ASI alters the SAE's behavior in directions corresponding to the dead subspace, we perform a comparative analysis between Attention Output SAEs trained with and without ASI under identical configurations (same number of features, $K$, and all other hyperparameters).

**Feature alignment with the dead subspace.** We compute the cosine similarity between each SAE feature and the dead subspace, restricting the analysis to alive features (which accounts for the difference in total counts). The distribution of cosine values is summarized in Table 10. Across all intervals, the ASI-initialized SAE exhibits a larger number of alive features, while both methods exhibit very few features that align closely with the dead subspace. This suggests two possible explanations: (i) features in the dead subspace have extremely small magnitude and provide insufficient signal for the SAE to learn, or (ii) the dead subspace does not contain meaningful standalone features, and only small components of features reside in this region.

Table 10: Distribution of cosine similarity between SAE features and the dead subspace (alive features only).

| Method | [0.0, 0.05) | [0.05, 0.1) | [0.1, 0.15) | [0.15, 1] |
|--------|-------------|-------------|-------------|-----------|
| TopK | 10176 | 189 | 6 | 0 |
| ASI | 24576 | 936 | 8 | 0 |

**Reconstruction error in the dead subspace.** We further project the reconstruction error onto the dead subspace to quantify the SAE's reconstruction performance on components lying in this region.

Table 11: Reconstruction error projected onto the dead subspace.

| Method | MSE in dead subspace |
|--------|---------------------|
| TopK | 0.00350 |
| ASI | 0.00334 |

As shown in Table 11, the reconstruction errors are nearly identical between the two methods, with the ASI showing only a marginal improvement.

Overall, these analyses indicate that ASI has minimal impact on the SAE's behavior within the dead subspace, while substantially reducing the number of dead features.

