# OpenReview forum: "Attention Layers Add Into Low-Dimensional Residual Subspaces"
_ICLR.cc/2026/Conference — Submitted to ICLR 2026_

### Official Review · Reviewer_TjHM · 2025-10-19

**Soundness:** 3
**Presentation:** 2
**Contribution:** 2
**Rating:** 4
**Confidence:** 4

**Summary:**

The authors demonstrate that the outputs of attention layers are relatively low-rank, with ~60% of dimensions accounting for 99% of variance. They use this to develop a new method for initializing SAEs which initializes the feature directions into this “active” space, rather than initializing uniformly across the entire activation space. They demonstrate that this significantly decreases the number of dead SAE features.

**Strengths:**

1. The claim that attention outputs are relatively low-rank is demonstrated convincingly and thoroughly.
2. I like how they follow up on the low-rank finding and investigate the source of this phenomenon in section 4.3.
3. The idea of initializing SAEs based on how active a given subspace is is novel and interesting, and I would expect it to help improve SAE training (in terms of not just dead features but also other things which the authors didn’t mention, e.g. reducing the number of training tokens needed)
4. The paper is mostly well-written, easy to follow, and the mathematical notation is clear (except for some of the figures).

**Weaknesses:**

1. Given that most activation spaces in LLMs aren’t low-rank, the usefulness of this technique is somewhat limited. As far as I’m aware, training SAEs on the attention output is somewhat uncommon, so if that is the only activation space where this technique is relevant, most real-world use cases would not benefit from this approach.
2. The authors do not study other activation spaces which *are* commonly used when training SAEs, e.g. the hidden activations in the MLP. I would really like to see many other spaces included in e.g. Figure 1\.
3. The definition of the initialization method itself is somewhat unclear. The authors say (around line 340\) that the decoder is initialized to span the active subspace, which makes sense but since there are significantly more decoder directions than dimensions in the active subspace, there are many ways that one could do this. As such, currently the central piece of the initialization method isn’t clearly explained in the main text (I only understood it by reading the pseudocode in appendix I).
4. Figure 7(b) is confusing, are they saying that their SAEs find up to 1M alive features in an 8B model? I would expect insanely high levels of feature splitting at that point; I think arguably having dead features is preferable to having all semantically meaningful features be arbitrarily split into many SAE features. This is making me confused about what the features here look like, and as far as I can tell the authors do not present any qualitative evidence on what type of features their SAEs learn (especially the ones with very high numbers of features).
5. SAEs trained with this initialization might end up missing certain features. Even though the active subspace contains the vast majority of the variance (99%), SAEs have *many* features and one would expect that roughly 1% of them should fall into the “dead” subspace. As such, I would somewhat expect that SAEs trained with this initialization method would fail to learn (some of) those features.
6. Many of the figures are difficult to interpret. E.g. in Figure 1, in the left subfigure and the right subfigure, it isn’t clear what model is being used.
7. Many of the figures in the appendices lack proper captions and are thus very difficult to interpret (e.g. Figure 15 — what model are you using, what layer etc).
8. The figures are clearly exported as raster images rather than vector images, and they get blurry when you zoom in. The authors really should export in a vector format instead (matplotlib makes exporting to pdf very easy).
9. Section 5.2 defines SVD a second time, after it was already defined in section 4.1. Arguably defining SVD is already slightly redundant for a technical audience; defining it twice is certainly redundant.
10. Typo in footnote 5 on page 6: “Another prominent open-source SAEs…”. Should be something like “Another prominent set of open-source SAEs…”

**Questions:**

1. ​​What does Figure 6 mean? TopK is an activation function, ASI is an initialization method, how are these comparable? I’m really confused by the labels here.
2. For most figures, can you clarify which models you’re using? Many of them currently do not include this.
3. In the initialization method, people have to choose the fraction of variance explained to use as a threshold. Do you have any advice on how to choose the right number here?

---

> ### Author Response · Authors · 2025-11-24
> **Response to Reviewer TjHM (1 / 6)**
>
> We sincerely appreciate your recognition of our work, with highlights in novelty, soundness and interest. We would also like to acknowledge the detailed and constructive feedback and questions provided that help strengthen our work, which we summarize as follows:
>
> - Will ASI miss some features in dead subspace?
> - Explanation for the scaling law experiment
> - The usefulness of our technique
> - The spectra of other activation types
> - Some detailed writing improvement
>
> All of these aspects are helpful and insightful. We sorted them in a reasonable logical order we think and respond in the following comments.

---

> ### Author Response · Authors · 2025-11-24
> **Will ASI miss some features in dead subspace? (2 / 6)**
>
> > (Reviewer TjHM) SAEs trained with this initialization might end up missing certain features. Even though the active subspace contains the vast majority of the variance (99%), SAEs have many features and one would expect that roughly 1% of them should fall into the “dead” subspace. As such, I would somewhat expect that SAEs trained with this initialization method would fail to learn (some of) those features.
>
> It is really a reasonable concern!
>
> To evaluate this, we **compared Attention Output SAEs trained with and without ASI** under identical sonfigurations (same number of features, same K, and all other hyperparameters held constant). Our analysis focused on two aspects:
>
> 1. **Number of features lying in the dead subspace**
>
> We compute the cosine similarity between each SAE feature and the dead subspace (calculated only over alive features, which accouts for the different total counts in the table below). The distributions are shown below:
>
> | Method | [0.0, 0.05) | [0.05, 0.1) | [0.1, 0.15) | [0.15, 1] |
> |--------|-------------|-------------|-------------|-----------|
> | TopK   | 10176       | 189         | 6           | 0         |
> | ASI    | **24576**       | **936**         | **8**           | 0         |
>
> **Across all intervals, ASI has a larger or equal number of alive features**.
>
> Moreover, even without ASI, few features fall into the dead subspace. We think this has two possible factors:
> (1) features in the dead subspace are extremely small and provide too little signal for the SAE to learn;
> (2) the dead subspace may not contain meaningful standalone features, and only tiny components of features reside there.
>
> 2. **Reconstruction error within the dead subspace**
>
> We further project the reconstruction error onto the dead subspace to quantify how well the SAE reconstructs this subspace. The results are:
>
> | Method | MSE in dead subspace |
> |--------|-----------------------|
> | TopK   | 0.00350               |
> | ASI    | **0.00334**               |
>
> **ASI achieves a lower reconstruct error in the dead subspace.**
>
> Although these analyses are relatively simple, they indicate that the influence of ASI on the SAE’s behavior within the dead subspace is minimal.
>
> **We have incorporated this part in Appendix O.2 in the revised manuscript.**

---

> ### Author Response · Authors · 2025-11-24
> **Explanation for the scaling law experiment (3 / 6)**
>
> > (Reviewer TjHM) Figure 7(b) is confusing, are they saying that their SAEs find up to 1M alive features in an 8B model? I would expect insanely high levels of feature splitting at that point; I think arguably having dead features is preferable to having all semantically meaningful features be arbitrarily split into many SAE features. This is making me confused about what the features here look like, and as far as I can tell the authors do not present any qualitative evidence on what type of features their SAEs learn (especially the ones with very high numbers of features).
>
> You are right! According to prior work [1], when training a very large SAE, both feature splitting and feature absorbing become increasingly severe. However, for Attention Output SAEs, **even the smallest SAE size used in our experiments (16,384 features, i.e., 4 \times d_{\text{model}}) still exhibits around 30% dead features**. **Dead features at such small scales—where feature splitting is not expected to occur—are what we aim to avoid**. The primary purpose of our method is to prevent dead features at model sizes where they should not naturally arise.
>
> Regarding Figure 7(b): **our intention is not to recommend training SAEs with extremely large numbers of features** (e.g., 1M). Instead, the figure **aims to demonstrate that our initialization strategy remains effective across a wide range of SAE sizes**. In all other experiments, we continue to use a reasonable number of features (32,768, i.e., 8 \times d_{\text{model}}), which aligns with common practice.
>
> As an additional note, we evaluated the monosemanticity of firing patterns from both the base TopK SAE and the ASI-enhanced SAE. Following the rubric of Cunningham et al. [2], we conducted a blinded assessment of 100 randomly sampled features and found no statistically significant differences between the methods at this analysis scale; any observed variation was marginal. Although this evaluation is relatively simple, our approach does not modify the SAE architecture or training objective, so the risk of influencing feature quality is small.
>
> **We have included additional details and discussion in Appendix O.1 of the revised manuscript.**
>
> [1] [A is for Absorption: Studying Feature Splitting and Absorption in Sparse Autoencoders](https://arxiv.org/abs/2409.14507)
>
> [2] https://transformer-circuits.pub/2024/june-update/index.html

---

> ### Author Response · Authors · 2025-11-24
> **The usefulness of our technique (4 / 6)**
>
> > (Reviewer TjHM) Given that most activation spaces in LLMs aren’t low-rank, the usefulness of this technique is somewhat limited. As far as I’m aware, training SAEs on the attention output is somewhat uncommon, so if that is the only activation space where this technique is relevant, most real-world use cases would not benefit from this approach.
>
> Thanks for your insightful question!
>
> It is true that most activation spaces in LLM aren't low rank. It won't apply on all training sites but **is critical to understanding attention behavior, which would otherwise be hard for existing training techniques**.
>
> Meanwhile, **we find that in some special models or datasets the residual streams also have low effective dimensions**. Such as feeding SlimPajama to Qwen3-4B and feeding pure code data to Qwen3-8B (details in Figure 9). Although the cause of this phenomenon is still unclear, **it provides some application scenarios for our technology beyond attention**.

---

> ### Author Response · Authors · 2025-11-24
> **The spectra of other activation types (5 / 6)**
>
> > (Reviewer TjHM) The authors do not study other activation spaces which are commonly used when training SAEs, e.g. the hidden activations in the MLP. I would really like to see many other spaces included in e.g. Figure 1.
>
> Thank you for your insightful suggestion!
>
> **We further compute the effective dimensions of the hidden activations of MLP and the concatenated outputs of all attention heads** across all layers and three models~(llama-3.1-8b, gemma-2-9b, qwen3-8b). We have incorporated these results in Appendix E.2 and Figure 10.

---

> ### Author Response · Authors · 2025-11-24
> **Some detailed writing improvement (6 / 6)**
>
> > (Reviewer TjHM) Many of the figures are difficult to interpret. E.g. in Figure 1, in the left subfigure and the right subfigure, it isn’t clear what model is being used.
>
> > (Reviewer TjHM) Many of the figures in the appendices lack proper captions and are thus very difficult to interpret (e.g. Figure 15 — what model are you using, what layer etc).
>
> > (Reviewer TjHM) The figures are clearly exported as raster images rather than vector images, and they get blurry when you zoom in. The authors really should export in a vector format instead (matplotlib makes exporting to pdf very easy).
>
> Thank you for pointing out these flaws!
>
> We have updated the caption of Figure 1 in the manuscript, making the dataset and model used clear.
>
> **In the section 4.2** of the revised manuscript, we summarize **the key information about the model and dataset used in Section 4**, and provide complete information in Appendix B.  **In the appendix F** of the revised manuscript, we provide **complete sae training details (include model, dataset and other hyperparameters)** used in Section 5.
> We have added appropriate captions to all the images in the revised manuscript and converted them into vector images.
>
>
> > (Reviewer TjHM) The definition of the initialization method itself is somewhat unclear. The authors say (around line 340) that the decoder is initialized to span the active subspace, which makes sense but since there are significantly more decoder directions than dimensions in the active subspace, there are many ways that one could do this. As such, currently the central piece of the initialization method isn’t clearly explained in the main text (I only understood it by reading the pseudocode in appendix I).
>
> Thanks for your detailed suggestions!
>
> In the revised manuscript, we changed $W_D \in span(V_{active})$ to the following more detailed description, consistent with the pseudocode in the appendix:
>
> We first randomly initialize the decoder weights $W_D \in \mathbb{R}^{h \times d}$ and then project their first $d_{\text{init}}$ columns onto the active subspace:
> $W_D \leftarrow W_D[:, :d_{\text{init}}] \ V_{\text{active}}^\top$
>
> > (Reviewer TjHM) Section 5.2 defines SVD a second time, after it was already defined in section 4.1. Arguably defining SVD is already slightly redundant for a technical audience; defining it twice is certainly redundant.
>
> > (Reviewer TjHM) Typo in footnote 5 on page 6: “Another prominent open-source SAEs…”. Should be something like “Another prominent set of open-source SAEs…”
>
> **We have simplified the description in section 5.2 and corrected the typo you pointed out in the revised manuscript**, and thank you again for your detailed review comments.

---

> > ### Comment · Reviewer_TjHM · 2025-11-25
> > **Response to rebuttals**
> >
> > Thank you for your detailed rebuttal! You’ve successfully resolved weaknesses 2, 3, 4, 6, 8, 9, and 10, and I’m increasing my score — well done :)
> >
> > Regarding the first point, I’d like to see the results in appendix O replicated on multiple models, multiple layers, ideally multiple hyperaparams etc. But I do feel reasonably convinced that this will probably generalize, so weakness 5 is partially resolved.
> >
> > Weakness 1 still stands, I agree that there are situations where you’re training on a relatively narrow dataset in which this becomes applicable in a broader range of subspaces, but as far as I know that’s fairly uncommon for SAEs.
> >
> > I also still think the figure captions should be more specific, e.g. Figures 14 and 15.

---

> ### Author Response · Authors · 2025-11-26
> **Response to Reviewer TjHM**
>
> Really thank you for taking the time to review our responses and the additional experiments. We greatly appreciate your feedback and support throughout the review process.
>
> Some additions:
> > (Reviewer TjHM) Weakness 1 still stands, I agree that there are situations where you’re training on a relatively narrow dataset in which this becomes applicable in a broader range of subspaces, but as far as I know that’s fairly uncommon for SAEs.
>
> You are right. After our searching, we only found four works (may miss some) related to training SAEs on the narrow dataset [1] [2] [3] [4]. [1] [2] [3] emphasizes that training SAEs on domain-specific datasets helps to find features in that domain with higher quality. [4] points out that training SAEs on the narrow dataset helps to improve reconstruction. Recent interpretability work from OpenAI [5] was also done on the code domain dataset (they trained a weight sparse model instead of SAEs). Although training SAEs on domain datasets is currently indeed fairly uncommon, we believe there is potential for further development, especially in specific application scenarios.
>
> Our initialization methods indeed have this weakness of its application scope. However, we believe that due to the other two findings—that the attention output shows a low-rank structure and that deads features are highly correlated with effective rank—this will not significantly affect the contribution and value of this paper.
>
> [1] [SAEs are highly dataset dependent: a case study on the refusal direction](https://www.alignmentforum.org/posts/rtp6n7Z23uJpEH7od/saes-are-highly-dataset-dependent-a-case-study-on-the)
>
> [2] [Domain-specific SAEs](https://www.lesswrong.com/posts/ojERTvdGWW6XRZAqr/domain-specific-saes)
>
> [3] [Resurrecting the Salmon: Rethinking Mechanistic Interpretability with Domain-Specific Sparse Autoencoders](https://www.arxiv.org/abs/2508.09363)
>
> [4] [Towards Atoms of Large Language Models](https://arxiv.org/abs/2509.20784)
>
> [5] [Weight-sparse transformers have interpretable circuits](https://cdn.openai.com/pdf/41df8f28-d4ef-43e9-aed2-823f9393e470/circuit-sparsity-paper.pdf)
>
> > (Reviewer TjHM) I also still think the figure captions should be more specific, e.g. Figures 14 and 15.
>
> Thank you for your feedback from a reader's perspective; it is very helpful for us to improve the presentation! We have updated the captions of Figures 13, 14, and 15 again, introducing the SAE architecture used and the activation position where the SAE was trained, and providing links to the more detailed descriptions in the appendix.

---

### Official Review · Reviewer_fThX · 2025-10-25

**Soundness:** 3
**Presentation:** 3
**Contribution:** 3
**Rating:** 6
**Confidence:** 4

**Summary:**

The paper studies the effective dimensionality of Transformer activations and finds that attention outputs occupy a low-dimensional subspace: roughly 60% of directions explain 99% of variance, whereas residual streams and MLP outputs are near full rank. The authors trace this low-rank structure mainly to the attention output projection matrix W_O. They show a strong correlation between low intrinsic dimension and dead features in SAEs. To address this, they introduce Active Subspace Initialization (ASI), which initializes SAE dictionaries in the top right-singular subspace of the target activations, sharply reducing dead features and improving reconstruction and downstream loss; the gains persist as the number of features scales, and combining ASI with SparseAdam further reduces dead features. They also show that ASI helps a sparse replacement model (Lorsa) by lowering dead parameters and improving reconstruction.

**Strengths:**

- The paper highlights a previously under-discussed phenomenon: attention outputs live in a lower-dimensional subspace (while this is not the case for MLP and residual activations).
- The connection to dead features in SAEs, while correlational, is supported by convincing evidence.
- The proposed initialization appears effective in practice.

**Weaknesses:**

- In Figure 4 the typical effective rank for MLPs is about 0.90–0.95, but the last layer shows a drop to roughly 65%. Any insights into what might cause this behavior? Did you apply your initialization to this layer as well, given it also shows a nontrivial fraction of dead features?
- Architectural implications of low-rank attention: If attention outputs are low rank, could one reduce per-head dimensionality while keeping d_model fixed, possibly increasing the number of heads, without losing much information? Is the observed low rank due to redundancy across heads or primarily to W_O anisotropy? A short discussion would help clarify whether narrower heads or different head counts would help or hurt.
- The paper does not discuss whether the initialization helps interpretability. Even a brief note or example on whether feature quality or concept alignment improves would be useful, though I understand this is not the main focus.

**Questions:**

See weaknesses.

---

> ### Author Response · Authors · 2025-11-23
> **Response to Reviewer fThX (1 / 4)**
>
> We sincerely appreciate your recognition of our work, with highlights in novelty, soundness and significance. We would also like to acknowledge the detailed and constructive feedback and questions provided that help strengthen our work, which we summarize as follows:
>
> - Further research on the MLP last layer
> - Architectural implications of low-rank attention output
> - Evaluate the interpretability for ASI features
>
> All of these aspects are helpful and insightful. We sort them in a reasonable logical order we think and respond in the following comments.

---

> ### Author Response · Authors · 2025-11-23
> **Further research on the MLP of last layer (2 / 4)**
>
> > (Reviewer fThX) In Figure 4 the typical effective rank for MLPs is about 0.90–0.95, but the last layer shows a drop to roughly 65%. Any insights into what might cause this behavior? Did you apply your initialization to this layer as well, given it also shows a nontrivial fraction of dead features?
>
> Thanks for your insightful question!
>
> You noticed that **MLP output of the last layer show a low effective rank** which we have not studied in depth.
>
> First, **we inspect other models** including qwen3-8b and gemma-2-9b, and we surprisingly **find that this phenomenon also occurs**. We think this might be a universal phenomenon in the models using this architecture (decoder-only transformer).
>
> We also visualize the singular value spectra of MLP output of the last layer. **Its singular value spectrum is generally lower in the latter half than that of other layers while keeping the same slope change(relatively flat), rather than dropping rapidly like attention output**(Figure 2).
>
> We apply our method (ASI) on this activation and successfully reduce the dead features while improving reconstruction, as shown in the following table.
>
> | **Metrics**            | **TopK** | **ASI** |
> |------------------------|---------:|--------:|
> | Proportion of Dead Features(%)   | 29.275     | **1.067**    |
> | Normalized MSE         | 0.10425  | **0.09001** |
>
> Unfortunately, we did not expore the further reasons of this phenomenon and left this part to future work. We think this does not affect the main contribution of this paper.

---

> ### Author Response · Authors · 2025-11-23
> **Architectural implications of low-rank attention output  (3 / 4)**
>
> > (Reviewer fThX) Architectural implications of low-rank attention: If attention outputs are low rank, could one reduce per-head dimensionality while keeping d_model fixed, possibly increasing the number of heads, without losing much information?
>
> Considering that attention outputs have a low-rank structure, a straightforward intuition to mitigate this might be increasing n_heads or d_heads. To investigate this, we examined the models pythia-2.8b and Qwen3-4B. Some relevant statistics are summarized below:
>
> | **Model**           | **pythia-2.8b** | **Qwen3-4B** |
> |---------------------|----------------:|-------------:|
> | **d_model**        | 2560            | 2560            |
> | **n_heads**        | 32              | 32            |
> | **d_head**         | 80              | **128**          |
> | **Effective Rank of Attention Output (mean across layers)** | **1766.1239** | 1240.1701 |
>
> Interestingly, Qwen3-4B, which has a larger d_head, exhibits a lower effective rank. Given the many other differences(eg. traning process) between these models, this comparison has a limited value, but it still suggests that **simply increasing d_head may not work**. Since open-source models have fixed configs, **we did not have the opportunity to fairly isolate the effect of a single hyperparameter.**
>
> Just a quick note, we think exploring approaches beyond hyperparameter tuning may also be worthwhile--for instance, constraining the overlap between the subspaces of different heads during training to reduce redundancy across heads.
>
> > (Reviewer fThX) Is the observed low rank due to redundancy across heads or primarily to W_O anisotropy?
>
> Mathematically, we find that the low-rank structure is structurally imposed by the anisotropy of the attention output projection matrix $W^O$. That is, as long as $W^O$ maintains this anisotropy, attention outputs will exhibit low-rank structure regardless of how each head’s output varies.
>
> However, we think this is a bidirectional problem:
> - the strong compression of variance along certain directions by $W^O$ weakens the corresponding gradient signals, preventing heads from learning information useful for subsequent layers. In this sense, the anisotropy of $W^O$ contributes to redundancy across heads.
> - At the same time, this redundancy primarily carries noisy or uninformative content, which $W^O$ cannot effectively leverage, thereby reinforcing the compression of these directions to limit their impact on downstream representations.
>
> Consequently, if we can really avoid redundancy across heads during training, this would likely also reduce the anisotropy in $W^O$.

---

> ### Author Response · Authors · 2025-11-23
> **Evaluate the interpretability for ASI features (4 / 4)**
>
> > The paper does not discuss whether the initialization helps interpretability. Even a brief note or example on whether feature quality or concept alignment improves would be useful, though I understand this is not the main focus.
>
> Thank you for raising this point.
>
> **We evaluate the degree of monosemanticity of the firing patterns of the features from base TopK SAE and ASI enhanced SAE**. In a blinded evaluation of 100 features using the rubric of Cunningham et al. [1], **we found no statistically significant differences** between experimental runs at this scale of analysis, with any observed variations being marginal. Although this comparison is somewhat simple, **our approach does not focus on feature quality and involve any change to the SAE architecture. The risk of affecting feature quality is relatively small.**
>
> **We have incorporated more details of this part in Appendix O.1 in the revised manuscript.**
>
> [1] https://transformer-circuits.pub/2024/june-update/index.html

---

### Official Review · Reviewer_GERd · 2025-11-03

**Soundness:** 3
**Presentation:** 2
**Contribution:** 3
**Rating:** 4
**Confidence:** 3

**Summary:**

The paper investigates the intrinsic dimensionality of attention-layer outputs, showing that these representations are highly redundant: roughly 60% of directions capture 99% of the variance. The authors propose a connection between the low-rank structure of attention outputs and the dead-feature phenomenon observed in sparse dictionary learning. To address this, they introduce a novel initialization scheme for self-attentive encoders (SAEs) that projects feature directions into the activation’s active subspace, with the aim of reducing dead features.

**Strengths:**

The paper develops principled methodologies for probing the intrinsic dimensionality of various network outputs (attention, MLPs, and residual streams), yielding useful diagnostics for how different components contribute to loss and representation capacity.

Establishing a link between the low-rank structure of attention outputs and the dead-feature problem is an insightful contribution; this hypothesis could deepen our understanding of representation collapse and suggest principled prevention strategies.

The proposed initialization strategy appears to alleviate the dead-feature issue for moderate token/model sizes, suggesting practical value for improving representational utilization without costly retraining.

**Weaknesses:**

The biggest issue I have with the paper is that the experimental section seems to be disorganized. It is hard to understand which models, datasets, and setups were used to prepare the experiments and support the claims. To be precise:

Section 4.1 studies the intrinsic dimensions of the attention outputs, MLP outputs, and residual streams. However, it is unclear which models (not just model families, but exact sizes) were used in the experiments. What does the x-axis in the “across layers” image mean? (Does x/6 mean that the value was measured after the first x×6 layers?) What parts/splits/number of tokens were used in the study?

Section 4.2 studies the spectra of the layers in the model. It also reports the impact of the principal components on the model’s loss. It seems that this experiment was conducted on a different dataset than the one from Section 4.1.

Section 5.1 / Figure 4: It is again unclear on which datasets the values were obtained. If the goal was to depict correlation, why not plot the dead feature proportions against the intrinsic dimension to actually measure the correlation?

The paper is compared only against the Top-K method and its variant with an additional loss (AuxK), which is not explained in the main text.

In general, the paper seems very interesting, but it requires a more rigorous and structured experimental design.

**Questions:**

I would suggest the authors unify the datasets and models across all the studies in the paper,  preferably adding also a section that summarises the used models and datasets before moving to the analysis of the results.

---

> ### Author Response · Authors · 2025-11-21
> **Response to Reviewer GERd (1 / 5)**
>
> We sincerely appreciate your recognition of our work, with highlights in novelty, soundness and significance. We would also like to acknowledge the detailed and constructive feedback and questions provided that help strengthen our work, which we summarize as follows:
>
> - Clarification and Rewrite for Section 4
> - Clarification and Rewrite for Section 5
> - A unified introduction to the dataset and model used
> - Rigorous and structured experimental design
>
> All of these aspects are helpful and insightful. We sorted them in a reasonable logical order we think and respond in the following comments.

---

> ### Author Response · Authors · 2025-11-21
> **Clarification and Rewrite for Section 4 (2 / 5)**
>
> > (Reviewer GERd) Section 4.1 studies the intrinsic dimensions of the attention outputs, MLP outputs, and residual streams. However, it is unclear which models (not just model families, but exact sizes) were used in the experiments.
>
>
> Sorry, this was exactly not clearly clarified.
>
> In figure 1, the left subfigure (across layer analysis) is conducted on **Llama-3.1-8B**; the middle subfigure (across layer analysis) is conducted on **Llama-3.1-8B, Gemma-2-9B, Qwen3-8B**; the right subfigure (across dataset) is conducted on **Llama-3.1-8B**.
>
> **We have updated the Figure 1 and its caption in the revised manuscript to provide detailed info of the model and dataset used.**
>
> > (Reviewer GERd) What does the x-axis in the “across layers” image mean? (Does x/6 mean that the value was measured after the first x×6 layers?)
>
> Sorry, this was not clearly clarified.
>
> The x-axis values represent the **Relative Layer Depth**. For example, for a model with 32 layers, layer 7 (index starting from 0) has a relative layer depth of $1/4$.
>
> We have updated Figure 1 in the revised manuscript to make this clearer.
>
> > (Reviewer GERd) What parts/splits/number of tokens were used in the study?
>
> We are willing to explain further.
>
> All experiments of **singular value decomposition** used 10M tokens, which are randomly sampled from the used dataset.
>
> We empirically verify that this sample size is sufficient to produce stable and reproducible singular spectrum analysis. Specifically, for the attention output of layer 15 of llama-3.1-8b, **we performed 5 times of singular value decompositions using different 10 million tokens for each**, and calculated the Coefficient of Variation (CV, std / mean) for each singular value across these 5 runs. The maximum CV was only 4.9e-3, and the mean and standard deviation of the effective rank computed from these 5 SVD results were 2523.165 and 0.404, respectively, with a CV of 1.5e-4. These error experiments show that using 10 million tokens for singular value decomposition is sufficiently stable.
>
> **We have incorporated the info of used tokens in section 4.2 and the stability analysis in Appendix C in the revised manuscript.**
>
>
> > (Reviewer GERd) Section 4.2 studies the spectra of the layers in the model. It also reports the impact of the principal components on the model’s loss. It seems that this experiment was conducted on a different dataset than the one from Section 4.1.
>
> Sorry, this was exactly not clearly clarified.
>
> The results of the spectra analysis provided in section 4.2 (change to section 4.3 in the revised manuscript) are run on **SlimPajama dataset, same as the dataset used in Section 4.1 (Figure 1 left mid)**.

---

> ### Author Response · Authors · 2025-11-21
> **Clarification and Rewrite for Section 5 (3 / 5)**
>
> > (Reviewer GERd) Section 5.1 / Figure 4: It is again unclear on which datasets the values were obtained. If the goal was to depict correlation, why not plot the dead feature proportions against the intrinsic dimension to actually measure the correlation?
>
> We are willing to explain further.
>
> Figure 4 evaluate the dead features of SAEs **from LlamaScope** [1], which **are trained on SlimPajama dataset**. We state that "we use the same framework and data as the original study to evaluate the LlamaScope SAEs" in section 5.1.
>
> Ploting the dead feature proportions against the effective dimension is really a good suggestion! **We have added such a subfigure to Figure 4** in the revised manuscript, which shows the correlation more clearly.
>
> [1] He et al. Llama Scope: [Extracting Millions of Features from Llama-3.1-8B with Sparse Autoencoders](https://arxiv.org/abs/2410.20526)
>
> (Reviewer GERd) The paper is compared only against the Top-K method and its variant with an additional loss (AuxK), which is not explained in the main text.
>
> We may have a wrong understanding of your question. If you mean that we have not explained AuxK clearly, **we give the complete discription of TopK and AuxK where its first appears** in the image(FIgure 5) and in the text (line 396) in the revised manuscript to make it clearer.
>
> Just a quick note, we further compare against the JumpRelu SAEs and its variant with auxiliary loss (which is called pre-act loss) in Appendix M. Not only the TopK and TopK variant.

---

> ### Author Response · Authors · 2025-11-21
> **A clearer introduction to the dataset and model used (4 / 5)**
>
> > (Reviewer GERd) I would suggest the authors unify the datasets and models across all the studies in the paper, preferably adding also a section that summarises the used models and datasets before moving to the analysis of the results.
>
> Really thank you for your insightful suggestion!
>
> **In the section 4.2** of the revised manuscript, we summarize **the key information about the model and dataset used in Section 4**, and provide complete information in Appendix B.
> **In the appendix F** of the revised manuscript, we provide **complete sae training details (include model, dataset and other hyperparameters)** used in Section 5.
>
> This adjustment is indeed much clearer.

---

> ### Author Response · Authors · 2025-11-21
> **Rigorous and structured experimental design (5 / 5)**
>
> > (Reviewer GERd) In general, the paper seems very interesting, but it requires a more rigorous and structured experimental design.
>
> **Our first submission** contains some **unclear experimental setup descriptions** and a few **experiments with limited scope**, which may make you suggest a more rigorous and structured experimental design.
>
> In our revised manuscript, **we introduce the experimental setup in specific sections** as mentioned in the previous response. Beyond this, **we expand the scope for the experiments that may require in order to ensure any conclusion is well supported**.
>
> Specifically
> - We valid the universality of "attention outputs have low effective dimension" **across layers, models and datasets**(section 4, appendix E).
> - We conduct a systematic analysis of the statement "low effective dimension correlates with dead features" **across activation type** (section 5.1) and **SAE hyperparameters** (appendix G).
> - We add **error analysis** for **SVD** (appendix D) and **SAE metrics** (figure 5, 6; Appendix H).
> - We conduct a **statistical significance test** for the statement "ASI outperforms other baseline" (Section I).
> - We also evaluate the superiority of ASI on other **layers, models, datasets** (Appendix J) and **SAE activation function** (Appendix M).

---

### Official Review · Reviewer_e5FR · 2025-11-04

**Soundness:** 1
**Presentation:** 1
**Contribution:** 2
**Rating:** 2
**Confidence:** 3

**Summary:**

The authors investigate the "dead latents/features" problem in Sparse AutoEncoders (SAEs), under the hypothesis that the low intrinsic dimensionality of model activations in various locations may be responsible for dead features. Empirical results in specific settings show that, under singular-value decomposition, attention head outputs require fewer singular values to reconstruct up to a given precision than do residual stream activations or MLP outputs, with SAEs trained on attention head outputs also exhibiting more dead features than SAEs trained on residual-stream activations or MLP outputs. The authors leverage this insight to propose a new method, "Active Subspace Initialization" (ASI), which initializes SAEs from singular vectors of activations. In a specific setting, ASI shows fewer dead features relative to baselines, with additional benefits such as lower mean-squared error and lower language modeling loss given ASI-reconstructed activations.

**Strengths:**

The overall hypothesis -- i.e., that low intrinsic dimensionality may be the cause of dead features -- is an interesting one that could carry substantive implications for SAE training and mechanistic interpretability more broadly. ASI is well-motivated and shows promise as a potential/partial solution to the dead feature problem.

**Weaknesses:**

The most important weaknesses of the paper are the lack of technical details regarding the problem formulation and experiments, and the severely limited range of experiments. In many cases, there is not enough technical information regarding what experiments have been performed to properly interpret results; and in those cases where experiments are clearly defined, the current results are insufficient to support most substantive claims. Key weaknesses by section are elaborated below (with missing critical information elaborated in the Questions section of the review).

Introduction:
- The authors claim that "low intrinsic dimensionality strongly correlates with the number of dead features". However, there is no systematic analysis of obvious confounders such as dictionary size or the setting of K (L0) with which (top-K) SAEs are trained.

Background (sec 2):
- Discussion of linear subspaces (sec 2.1) focuses on the ill-defined notion of "low-rankness within self-attention mechanisms"; but there is in fact long line of work studying the phenomenon of linear subspace representation beyond attention mechanisms -- see, e.g., [1-5]. The subsequent discussion of the "linear representation hypothesis" in sec 2.2 focuses on the narrow interpretation of this hypothesis regarding 1-dimensional subspaces, when higher-dimensional subspaces (as studied in [1-5, inter alia]) would be a more relevant point of comparison for the intended study of intrinsic dimensionality.
- The authors state that "the linear representation hypothesis... has been validated across diverse model scales, architectures, and modalities"; but the cited works do not (to my knowledge) conduct any specific hypothesis testing. Instead, they train SAEs across such contexts, and demonstrate their utility in interpretability; but the linear representation hypothesis remains very much in contention -- see, e.g., [5, 6].

[1] Mikolov, T., Yih, W. T., & Zweig, G. (2013, June). Linguistic regularities in continuous space word representations. In Proceedings of the 2013 conference of the north american chapter of the association for computational linguistics: Human language technologies (pp. 746-751).
[2] Pennington, J., Socher, R., & Manning, C. D. (2014, October). Glove: Global vectors for word representation. In Proceedings of the 2014 conference on empirical methods in natural language processing (EMNLP) (pp. 1532-1543).
[3] Bolukbasi, T., Chang, K. W., Zou, J. Y., Saligrama, V., & Kalai, A. T. (2016). Man is to computer programmer as woman is to homemaker? debiasing word embeddings. Advances in neural information processing systems, 29.
[4] Vargas, F., & Cotterell, R. (2020, November). Exploring the Linear Subspace Hypothesis in Gender Bias Mitigation. In Proceedings of the 2020 Conference on Empirical Methods in Natural Language Processing (EMNLP) (pp. 2902-2913).
[5] Makelov, A., Lange, G., Geiger, A., & Nanda, N. (2024). Is This the Subspace You Are Looking for? An Interpretability Illusion for Subspace Activation Patching. In The Twelfth International Conference on Learning Representations.
[6] Sharkey, L., Chughtai, B., Batson, J., Lindsey, J., Wu, J., Bushnaq, L., ... & McGrath, T. (2025). Open problems in mechanistic interpretability. arXiv preprint arXiv:2501.16496.

Sec 3:
- The authors state that "This formulation shows that O can be viewed as the sum of low-dimensional outputs from each head." However, it shows only that there exists a linear transformation from Z to O, which may be full-rank. (Note that the authors use "rank" and "dimension" interchangeably; so I am interpreting the use of "low-dimensional" in the context of a linear transformation to mean "low-rank" -- please correct this interpretation if I am wrong.)

Sec 4:
- Throughout the entire paper, the authors refer to "intrinsic dimension", which they define here as the smallest integer k such that the ratio of summed squared singular values to total summed squared singular values exceeds a threshold. This definition of "intrinsic dimension" is nonstandard -- e.g., the notion of "intrinsic dimension" with which I am familiar, and have seen studied in the context of deep learning representations (see, e.g., sec 2.2 of [1] for an overview), is that of [2] -- and in the only citations of "intrinsic dimension" in the paper (Guth et al., 2023 and Staats et al., 2025), I do not see any mention of the term "intrinsic dimension".
    - Beyond the lack of clarity and consistency with respect to the literature introduced by this term, it also implies that the authors' proposed re-definition of intrinsic dimension does not carry the same weight as it would elsewhere. Why should we care about this particular measurement? Introducing a new measurement that is so central to the empirical findings requires theoretical or empirical justification regarding its significance, which is not provided.
- In sec 4, there is no systematic comparison between different layers, models, or SAE hyperparameters (e.g., dictionary size, train-time L0 (i.e., K in top-K), alpha as the auxiliary loss coefficient used to mitigate dead features, etc.).
    - Across layers and models, there is only a single result in Figure 1 that is not adequately explained (see the relevant point in the Questions section of the review, below).
    - Across hyperparameter settings: after consulting the full main paper and appendix, I cannot find the hyperparameter settings for the results reported in sec 4. (I see only tau in footnote 3 and some values reported in Appendix G, but it is not clear whether Appendix G applies to all experiments or only those in sec 5.) Even if these values were properly reported, it is crucial to also report results across hyperparameters to determine the extent to which the phenomena in question (i.e., the prevalence of dead features) are simply due to a specific hyperparameter setting.

[1] Janapati, S., & Ji, Y. (2024). A Comparative Study of Learning Paradigms in Large Language Models via Intrinsic Dimension. arXiv preprint arXiv:2412.06245.
[2] Facco, E., d’Errico, M., Rodriguez, A., & Laio, A. (2017). Estimating the intrinsic dimension of datasets by a minimal neighborhood information. Scientific reports, 7(1), 12140.

Sec 5:
- As in sec 4, there is no systematic comparison across layers, models, or hyperparameters (see the Questions section of the review, below).
- In sec 5.2 (and 5.4), the differences between AuxK and ASI seem quite small, and could easily be the result of random initialization/sampling, hyperparameter selection, or the specific model/layer/dataset considered. (The experiments below might place a substantial burden in terms of compute requirements, and if necessary could accordingly be run at a a smaller scale than the main-paper results.)
    - Regarding random initialization: experiments should be repeated with different random seeds, with error bars shown and statistical significance reported. (Note that this may or may not be relevant for ASI, which unlike the baselines is not randomly initialized; but it might still be relevant if training examples are shuffled. I suggest shuffling and running all experiments in Figure 5 and 6 with multiple random seeds.)
    - Regarding hyperparameter selection: as, per Appendix G.5, it appears that authors have already experimented across  batch size and learning rate. At minimum, I would suggest (a) reporting the results in these settings and indicating whether and why (not) the main-paper findings hold across these settings, and (b) additionally running experiments across dictionary size and K (and ideally alpha as well).
    - Model/layer/dataset: only a single model (Llama-3.1-8B), layer (15), and dataset (SlimPajama) are considered. All of the claims regarding the apparent superiority of ASI relative to baselines must be validated across multiple models, layers, and datasets. (In my opinion, given the setting, experimenting across a wider range of layers seems more important than doing so with respect to datasets, and datasets more important than models; but failing to consider multiple values for *any* of these experimental considerations is a clear red flag for cherrypicking.)

**Questions:**

Sec 4:
- What is the intended meaning and significance of the nonstandard "intrinsic dimension" measurement? (See the Weaknesses section of the review for discussion on related work studying "intrinsic dimension" -- why opt for this definition instead?)
- What is the x-axis in the leftmost plot of Figure 1?
- In sec 4.1, the authors state that "We empirically verify that this sample size is sufficient to produce stable and reproducible singular spectrum analysis." Where is the evidence to support this statement?
- In sec 4.2,
    - What, precisely, is fig 2a visualizing? (Please provide a mathematical specification in terms of objects defined in the paper.) What does it mean that "only 74.7% singular values exceed 1% of the maximum in attention outputs, versus 100.0% for MLP outputs and residual streams", and how/why is it relevant to the hypothesis under investigation?
    - In fig 2b, what does "fraction loss recovered" mean? (Again, please provide a formal specification.) Is this the train-time loss of SAEs, test-time loss, or language modeling loss of the underlying model? And what does "recovery" mean in this sense -- is it something to be minimized or maximized, and what would perfect recovery indicate?
- In sec 4.3,
    - What is "relative value" on the y-axis of Figure 3? And the authors state that, shown in this figure, "We compute and visualize both quantities across a set of directions aligned with the right singular vectors of attention output" -- what, specifically, are the two quantities and the set of directions? (Please provide formal specifications in response to both questions.)
    - The authors state that "Our analysis reveals that the low-rank structure of attention outputs O arises primarily from the anisotropy of W^O, which heavily compresses the output space into a lower-dimensional subspace" -- how, precisely, is anisotropy defined here, and what is the threshold or standard of evidence for arguing "heavily compresses"? What is the justification for this threshold/standard, and (where) is supporting evidence provided?
    - If I am reading the MHSA equation here correctly -- that is to say, that the dimensionality of the *union* of subspaces corresponding to each head is less than or equal to that of the *sum* -- isn't this completely trivial, as it would be guaranteed given any parameterization of attention heads?
        - Does "head_i" denote the output of attention heads, or something else?
        - Why does this matter (i.e., in what sense is it nontrivial), and how does it relate to the empirical results reported in section 4.3?

Sec 5:
- Are all SAEs (in Sec 5.2-5.4) trained on outputs of attention heads, MLPs, or the residual stream? (Appendix B.2 seems to suggest that main-paper results might be from attention head outputs, but this is never explicitly states.)
- In sec 5.2,
    - For fig 5, what is being visualized? in particular, it seems the x-axis is being described as "different subspace dimensions for activations with a full space dimension of 4096" -- what does this mean?
    - For the experiments whose results are reported in figs 5 and 6: what are the dictionary size and train-time K/L0 (for all methods), and alpha (for AuxK)?
        - And what is L0 on the y-axis of Figure 6? Does this refer to different values of K at train- or test-time? In the former case, are different SAEs trained for each value of K; or, in the latter case, is only one SAE trained at a single value of K then tested at different values of K (which would be out of distribution)?
        - (Note: "AuxK" is never defined -- I had to jump around in the document to find other uses of the term "auxiliary" and trace it to the preliminaries. I would state it and cite the relevant works here as well.)

---

> ### Author Response · Authors · 2025-11-20
> **Response to Reviewer e5FR (1 / 12)**
>
> We sincerely appreciate your review of our work, with highlights in the significance of our core findings and the promise of our proposed approach. We would like to acknowledge the thorough and constructive feedback and questions provided which help strengthen our work, which we summarize as follows:
> - Correction of the Misused Term "Intrinsic Dimension"
> - Clarification and Rewrite for Section 4
> - Clarification and Rewrite for Section 5
> - Systematic Analysis of the Claim "Low Effective Dimension Correlates with Dead Features"
> - Error Analysis
> - Statistical Significance Test
> - Evaluate ASI across Different Hyperparameters
> - Evaluate ASI across Different Layers, Models, Datasets
> - Clarification and Rewrite for Section 2, 3
>
> All of these aspects are helpful and insightful. We sorted them in a reasonable logical order we think and respond in the following comments.

---

> ### Author Response · Authors · 2025-11-20
> **Correction of the Misused Term "Intrinsic Dimension" (2 / 12)**
>
> > (Reviewer e5FR) Throughout the entire paper, the authors refer to "intrinsic dimension", which they define here as the smallest integer k such that the ratio of summed squared singular values to total summed squared singular values exceeds a threshold. This definition of "intrinsic dimension" is nonstandard -- e.g., the notion of "intrinsic dimension" with which I am familiar, and have seen studied in the context of deep learning representations (see, e.g., sec 2.2 of [1] for an overview), is that of [2] -- and in the only citations of "intrinsic dimension" in the paper (Guth et al., 2023 and Staats et al., 2025), I do not see any mention of the term "intrinsic dimension".
>
> > (Reviewer e5FR) Beyond the lack of clarity and consistency with respect to the literature introduced by this term, it also implies that the authors' proposed re-definition of intrinsic dimension does not carry the same weight as it would elsewhere. Why should we care about this particular measurement? Introducing a new measurement that is so central to the empirical findings requires theoretical or empirical justification regarding its significance, which is not provided.
>
> Thank you very much for your insightful comment. **The term “Intrinsic Dimension” was indeed misused in our paper, and its definition was not standard.**
>
> Our goal was to **quantify the effective dimension of activations** in order to compare different type of activations. After further investigation, we have revised this term to **effective rank**, following the standardized definition from Roy et al.[1]. **We have recalculated this metric for all relevant results** in the revised manuscript.
> Despite this correction, **the experimental results do not affect the conclusions** in the paper, including:
> - “Attention outputs consistently display the strongest low-rank structure compared to MLP outputs and residual streams.”
> - “Low effective dimensionality strongly correlates with the number of dead features.”
>
> To substantiate these conclusions, we provide more systematic experiments in the following response.
>
> [1] Olivier Roy and Martin Vetterli. [The effective rank: A measure of effective dimensionality](https://ieeexplore.ieee.org/document/7098875)

---

> ### Author Response · Authors · 2025-11-20
> **Clarification and Rewrite for Section 4, Part 1 (3 / 12)**
>
> We will answer your questions one by one and explain what is updates in the revised manuscript.
>
> > (Reviewer e5FR) What is the x-axis in the leftmost plot of Figure 1?
>
> Sorry, this was not clearly clarified. This result is across layers, and the x-axis values represent the **Relative Layer Depth**. For example, for a model with 32 layers, layer 7 (index starting from 0) has a relative layer depth of $1/4$. **We have updated Figure 1 in the revised manuscript** to make this clearer.
>
> > (Reviewer e5FR) What, precisely, is fig 2a visualizing? (Please provide a mathematical specification in terms of objects defined in the paper.) What does it mean that "only 74.7% singular values exceed 1% of the maximum in attention outputs, versus 100.0% for MLP outputs and residual streams", and how/why is it relevant to the hypothesis under investigation?
>
> Sorry, this was not clearly clarified. Fig. 2a visualizes **the distribution of relative singular values**. Given a singular value vector $sv \in \mathbb{R}^n$, sorted in descending order, the visualization shows the relative singular value sv / sv[0] This figure demonstrates that the singular value spectrum of attention outputs decays faster compared to the other two activations. **We have updated Figure 2 in the revised manuscript** to make this clearer.
>
> "only 74.7% singular values exceed 1% of the maximum in attention outputs, versus 100.0% for MLP outputs and residual streams" means that if we set 1% of the maximum sigular value as a threshold, attention output has quite small variance in 25.3% dimensions. **This supports our conclusion** "Attention outputs display the strongest low-rank structure".
>
> > (Reviewer e5FR) In fig 2b, what does "fraction loss recovered" mean? (Again, please provide a formal specification.) Is this the train-time loss of SAEs, test-time loss, or language modeling loss of the underlying model? And what does "recovery" mean in this sense -- is it something to be minimized or maximized, and what would perfect recovery indicate?
>
> Sorry, this was not clearly clarified. "Fraction loss recovered" refers to **taking the language model loss with activations at a specific position ablated to zero as a baseline and reporting the fraction of that loss recovered as some components of that activation are reintroduced.** Following the approach in [1] and [2].
>
> Formal specification: given the original language model cross-entropy loss is $loss_{original}$, the loss after ablating the activation at a specific position to zero is $loss_{zero}$, and the loss after replacing the original activation projected to the subspace spanned by first $n$ singular vectors is $loss_{recovered}$. Then, for these $n$ components, the fraction loss recovered is calculated as:
>
> $\frac{loss_{zero} - loss_{recovered}}{loss_{zero} - loss_{original}}$
>
> **We have incorporated these in Sections 4.1 and Appendix C of the revised manuscript** for further clarity and elaboration.
>
> For attention output, only 39.1\% of the dimensions are needed to recover 99\% of the downstream loss (compared to 95.3\% and 96.9\% of the dimensions for MLP outputs and residual streams to recover the same proportion). **This indicates that the attention output's impact is largely concentrated in a small subspace.**
>
> "perfect recovery" means $loss_{recovered}$ = $loss_{original}$
>
> [1] [Towards monosemanticity: Decomposing language models with dictionary learning.](https://transformer-circuits.pub/2023/monosemantic-features/index.html)
>
> [2] [Improving dictionary learning with gated sparse autoencoders.](https://arxiv.org/abs/2404.16014)

---

> ### Author Response · Authors · 2025-11-20
> **Clarification and Rewrite for Section 4, Part 2 (4 / 12)**
>
> > (Reviewer e5FR) What is "relative value" on the y-axis of Figure 3? And the authors state that, shown in this figure, "We compute and visualize both quantities across a set of directions aligned with the right singular vectors of attention output" -- what, specifically, are the two quantities and the set of directions? (Please provide formal specifications in response to both questions.)
>
> Sorry, this was not clearly clarified. In "We compute and visualize both quantities across a set of directions aligned with the right singular vectors of attention output":
> - The both quantities refer to contribution of $Z$ and contribution of $W^O$. We state that "We refer to $Var(Z \hat{v})$ as the contribution of Z, capturing how much variance the concatenated head output $Z$ provides in that direction, and $\|v\|_2^2$ as the contribution of $W^O$, measuring how much the output projection $W^O$ scales or suppresses that direction." in line 295.
> - The set of directions refers to **the directions defined by the right singular vectors of the activation matrix** after performing SVD. For an activation matrix $\widetilde{A} \in \mathbb{R}^{n \times d}$, $\widetilde{A} = U \Sigma V^\top$, the set of directions is represented by the column vectors of $V$.
>
> The red curve "attention output $O$" represents the standard deviation of $O$ along each direction of the "right singular vectors," which corresponds to the singular values.
>
> "Relative value" on the y-axis of Figure 3 originates from that **each y-value is the square root of the value after dividing it by the average of all values in the curve**, as stated in the caption: "all values are normalized to a common scale." The purpose of this normalization is to **align the three curves on the same scale** and to make the shape of the attention output $O$ 's curve correspond to the relative singular value curve previously shown.
>
> **We have updated this part in Section 4.4 and the caption of Figure 3 in the revised manuscript** to make this clearer.
>
> > (Reviewer e5FR) The authors state that "Our analysis reveals that the low-rank structure of attention outputs O arises primarily from the anisotropy of W^O, which heavily compresses the output space into a lower-dimensional subspace" -- how, precisely, is anisotropy defined here, and what is the threshold or standard of evidence for arguing "heavily compresses"? What is the justification for this threshold/standard, and (where) is supporting evidence provided?
>
> We are willing to explain further.
>
> We first prove that "the contribution of $W^O$ , measuring how much the output projection $W^O$ scales or suppresses that direction." Anisotropy refers to the varying scaling or suppression of $W^O$ across different directions.
>
> The justification for this is provided in Figure 3 and its caption. Specifically, in caption we state "The curve of $O$ closely follows that of $Z$ for the top components, whereas the downward trend of attention output at the tail is mainly due to $W^O$ 's contribution."
>
> > (Reviewer e5FR) If I am reading the MHSA equation here correctly -- that is to say, that the dimensionality of the union of subspaces corresponding to each head is less than or equal to that of the sum -- isn't this completely trivial, as it would be guaranteed given any parameterization of attention heads?
> > - Does "head_i" denote the output of attention heads, or something else?
> > - Why does this matter (i.e., in what sense is it nontrivial), and how does it relate to the empirical results reported in section 4.3?
>
> We are willing to explain further.
>
> $head_i$ denotes the output of a single attention head(index i).
>
> You are correct in pointing out that **the MHSA equation is trivially true in its mathematical formulation.** The goal is to **provide an intuitive understanding** of the cause behind the phenomenon observed in our paper: in the standard MHSA implementation, **attention output residing in a low-dimensional subspace will inevitably occur as long as the spaces of each attention head's output overlap.**

---

> ### Author Response · Authors · 2025-11-20
> **Clarification and Rewrite for Section 4, Part 3 (5 / 12)**
>
> > (Reviewer e5FR) Across hyperparameter settings: after consulting the full main paper and appendix, I cannot find the hyperparameter settings for the results reported in sec 4. (I see only tau in footnote 3 and some values reported in Appendix G, but it is not clear whether Appendix G applies to all experiments or only those in sec 5.) Even if these values were properly reported, it is crucial to also report results across hyperparameters to determine the extent to which the phenomena in question (i.e., the prevalence of dead features) are simply due to a specific hyperparameter setting.
>
> For all experiments in Section 4:
> -  SVD, effective rank , contribution of $Z$ , contribution of $W^O$ : These results are calculated directly using the internal activations from the language model without any other hyperparameters. The activation sources used in Section 4 are provided in Section 4.2 and Appendix B, which includes details on the models, datasets, and activation positions. The number of tokens used is 10 million.
> - Fraction loss recovered : In all cases, the cross-entropy loss for the language model is computed on the same 4M tokens.
>
> **In the section 4.2** of the revised manuscript, we summarize **the key information about the model, dataset and the num of tokens used in Section 4**, and provide complete information in Appendix B.
>
> The phenomenon "Attention outputs consistently display the strongest low-rank structure compared to MLP outputs and residual streams." **is validated across layers, models, datasets**(Figure 1).
>
> As for the phenomenon "the prevalence of dead features", **we provide more evidence across different SAE hyperparameters (L0, dictionary size), models, layers, datasets in the following response.**

---

> ### Author Response · Authors · 2025-11-20
> **Clarification and Rewrite for Section 5 (6 / 12)**
>
> > (Reviewer e5FR) Are all SAEs (in Sec 5.2-5.4) trained on outputs of attention heads, MLPs, or the residual stream? (Appendix B.2 seems to suggest that main-paper results might be from attention head outputs, but this is never explicitly states.)
>
> All SAEs in Section 5.2-5.4 are trained on attention output.
>
> **We explain this in Section 5.2 and provide more SAE training details in Appendix F in the revised manuscript**.
>
> > (Reviewer e5FR) For fig 5, what is being visualized? in particular, it seems the x-axis is being described as "different subspace dimensions for activations with a full space dimension of 4096" -- what does this mean?
>
> We are willing to explain further.
>
> Figure 5 visualizes some SAE metrics of using ASI with different $d_{init}$.  The phrase "different subspace dimensions" refers to different $d_{init}$. **We update this phrase in the caption of Figure 5 in revised manuscript** to make this clearer.
>
> Emphasize "activations with a full space dimension of 4096" is to tell readers the rightest point($d_{init}=4096$) in each figure is the result of SAE without ASI. We state in Sec 5.2 that "As $d_{init}$ decreases from the full space dimension within a certain range, the number of dead features in the SAE rapidly drops, with a corresponding improvement in Delta LM loss (Figure 5)." "Setting $d_{init}$ equal to the full space dimension is equivalent to not using Active Subspace Initialization"
>
> > (Reviewer e5FR) For the experiments whose results are reported in figs 5 and 6: what are the dictionary size and train-time K/L0 (for all methods), and alpha (for AuxK)?
>
> Figure 5: dictionary size is 32768; train-time L0 is 50.
>
> Figure 6: dictionary size is 32768; train-time L0 is 50; alpha is 1/32 following Gao et al[1].
>
> **We provide the complete SAE training details in Appendix F in the revised manuscript.**
>
> [1] Leo Gao, Tom Dupré la Tou ... [Scaling and evaluating sparse autoencoders](https://arxiv.org/abs/2406.04093)
>
> > (Reviewer e5FR) And what is L0 on the y-axis of Figure 6? Does this refer to different values of K at train- or test-time? In the former case, are different SAEs trained for each value of K; or, in the latter case, is only one SAE trained at a single value of K then tested at different values of K (which would be out of distribution)?
> (Note: "AuxK" is never defined -- I had to jump around in the document to find other uses of the term "auxiliary" and trace it to the preliminaries. I would state it and cite the relevant works here as well.)
>
> It is the train-time L0. **SAEs are trained using different L0(K).**
>
> Auxk means **training Topk SAEs enhanced with auxiliary loss**. In line 396 we state "It achieves near-zero dead features and slightly superior results compared to the auxiliary loss approach (AuxK)"

---

> ### Author Response · Authors · 2025-11-20
> **Systematic analysis of the claim "low effective dimension correlates with dead features" (7 / 12)**
>
> > (Reviewer e5FR) The authors claim that "low intrinsic dimensionality strongly correlates with the number of dead features". However, there is no systematic analysis of obvious confounders such as dictionary size or the setting of K (L0) with which (top-K) SAEs are trained.
>
> > (Reviewer e5FR) It is crucial to also report results across hyperparameters to determine the extent to which the phenomena in question (i.e., the prevalence of dead features) are simply due to a specific hyperparameter setting.
>
> We appreciate these insightful questions. This is indeed a reasonable and thought-provoking concern.
>
> The statement “low effective dimensionality strongly correlates with the number of dead features” is primarily supported by Figure 4 in the paper. In that figure, **we evaluate the effective rank of the residual stream, attention output, and MLP output at every layer of Llama-3.1-8B, along with the proportion of dead features in the SAEs trained on them**. All SAEs are taken from Llamascope[1], which has the same feature dimensionality (32768) and sparsity level ($L_0$ = 50).
>
> In light of your concern, **we have additionally trained and included results across different dictionary sizes** (feature nums = 16384, 32768, 65536) and different sparsity levels ($L_0$ = 32, 64, 128). The table below reports the proportion of dead features under these various settings. SAEs are trained on Llama-3.1-8B, which you can refer to Figure 4 for the effective rank of each activation. As shown, for any given configuration, the attention output—having lower effective rank—consistently exhibits a higher number of dead features.
>
> #### L0=32
> | activation type (effective rank) \ feature nums       | 16384   | 32768   | 65536   |
> |----------------------------------------|---------|---------|---------|
> | **Layer7 attention output (2351)**     | **84.80%** | **90.31%** | **94.02%** |
> | Layer7 residual stream (3664)          | 1.61%   | 6.69%   | 17.10%  |
> | **Layer15 attention output (2506)**    | **68.70%** | **79.86%** | **87.22%** |
> | Layer15 residual stream (3611)         | 27.01%  | 45.70%  | 61.33%  |
> | **Layer23 attention output (2654)**    | **58.45%** | **70.48%** | **78.81%** |
> | Layer23 residual stream (3634)         | 0.13%   | 0.26%   | 1.35%   |
>
> #### L0=64
> | activation type (effective rank) \ feature nums       | 16384   | 32768   | 65536   |
> |----------------------------------------|---------|---------|---------|
> | **Layer7 attention output (2351)**     | **66.97%** | **75.65%** | **82.96%** |
> | Layer7 residual stream (3664)          | 0.02%   | 0.06%   | 0.15%   |
> | **Layer15 attention output (2506)**    | **41.65%** | **54.68%** | **67.43%** |
> | Layer15 residual stream (3611)         | 1.73%   | 7.96%   | 18.83%  |
> | **Layer23 attention output (2654)**    | **41.58%** | **56.70%** | **67.05%** |
> | Layer23 residual stream (3634)         | 0.15%   | 0.09%   | 0.08%   |
>
> #### L0=128
> | activation type (effective rank) \ feature nums       | 16384   | 32768   | 65536   |
> |----------------------------------------|---------|---------|---------|
> | **Layer7 attention output (2351)**     | **49.85%** | **56.96%** | **64.83%** |
> | Layer7 residual stream (3664)          | 0.00%   | 0.00%   | 0.02%   |
> | **Layer15 attention output (2506)**    | **15.09%** | **25.61%** | **37.11%** |
> | Layer15 residual stream (3611)         | 0.07%   | 0.12%   | 0.44%   |
> | **Layer23 attention output (2654)**    | **21.90%** | **39.02%** | **52.68%** |
> | Layer23 residual stream (3634)         | 0.14%   | 0.09%   | 0.07%   |
>
> **We have incorporated these updates in Appendix G of the revised manuscript.**
>
> [1] He et al. [Llama Scope: Extracting Millions of Features from Llama-3.1-8B with Sparse Autoencoders](https://arxiv.org/abs/2410.20526)

---

> ### Author Response · Authors · 2025-11-20
> **Error analysis (8 / 12)**
>
> ## Singular Value Decomposition
> > (Reviewer e5FR) In sec 4.1, the authors state that "We empirically verify that this sample size is sufficient to produce stable and reproducible singular spectrum analysis." Where is the evidence to support this statement?
>
> We have added the following experiment to demonstrate this.
>
> For the attention output of layer 15 of llama-3.1-8b, **we performed 5 times of singular value decompositions using different 10 million tokens for each**, and calculated the Coefficient of Variation (CV) for each singular value across these 5 runs. The maximum CV was only 4.9e-3, and the mean and standard deviation of the effective rank computed from these 5 SVD results were 2523.165 and 0.404, respectively, with a CV of 1.5e-4.
>
> These error experiments show that using 10 million tokens for singular value decomposition is sufficiently stable.
>
> **We have incorporated these updates in Appendix D of the revised manuscript.**
>
> ## SAE Metrics
> > (Reviewer e5FR) Regarding random initialization: experiments should be repeated with different random seeds, with error bars shown and statistical significance reported. (Note that this may or may not be relevant for ASI, which unlike the baselines is not randomly initialized; but it might still be relevant if training examples are shuffled. I suggest shuffling and running all experiments in Figure 5 and 6 with multiple random seeds.)
>
> **We use 3 different random seeds for all experiments in Figure 5 and Figure 6** and compute the mean values and the standard deviations of each metrics
>
> #### Figure 5
> | Dinit | 64                | 128               | 256                | 512                | 1024               | 2048               | 3072               | 4096               |
> |--------|------------------|-----------------|------------------|------------------|------------------|------------------|------------------|------------------|
> | Dead   | 15432.67 ± 92.42 | 1803.67 ± 69.00 | 33.33 ± 4.51 | 5.00 ± 3.61  | **3.33 ± 2.08**      | 211.33 ± 18.56   | 4390 ± 83.02     | 16144.33 ± 129.52|
> | Normalized MSE | 0.33169 ± 0.00019 | 0.31648 ± 0.00014 | **0.31224 ± 0.00589** | 0.31649 ± 0.00003 | 0.31688 ± 0.00011 | 0.31698 ± 0.00005 | 0.32122 ± 0.00009 | 0.33367 ± 0.00014 |
> | Delta LM Loss (×1e-3) | 8.341 ± 0.060 | 7.864 ± 0.045 | **7.772 ± 0.062** | 7.834 ± 0.029 | 7.866 ± 0.049 | 7.860 ± 0.010 | 7.972 ± 0.021 | 8.375 ± 0.029 |
>
> #### Figure 6
> | L0 | Metric            | Base              | AuxK              | ASI               |
> |----|-----------------|-----------------|-----------------|-----------------|
> | 40 | Dead Feature     | 20395.00 ± 72.77 | 37.00 ± 5.20    | **10.67 ± 1.15**|
> |    | Normalized MSE   | 0.36323 ± 0.00018 | 0.34328 ± 0.00011 | **0.33724 ± 0.00022**|
> |    | Delta LM Loss (x1e-3) | 9.650 ± 0.040 | 8.801 ± 0.053  | **8.697 ± 0.033**|
> | 50 | Dead Feature     | 16144.33 ± 129.52 | 54.67 ± 5.51    | **4.00 ± 1.73** |
> |    | Normalized MSE   | 0.33367 ± 0.00014 | 0.32241 ± 0.00020 | **0.31680 ± 0.00006**|
> |    | Delta LM Loss (x1e-3) | 8.375 ± 0.029 | 7.958 ± 0.032  | **7.847 ± 0.050**|
> | 60 | Dead Feature     | 12239.33 ± 165.51 | 76.00 ± 9.17    | **2.67 ± 1.53** |
> |    | Normalized MSE   | 0.31106 ± 0.00007 | 0.30555 ± 0.00007 | **0.30000 ± 0.00005**|
> |    | Delta LM Loss (x1e-3) | 7.503 ± 0.059 | 7.325 ± 0.017  | **7.214 ± 0.026**|
> | 70 | Dead Feature     | 8854.67 ± 40.51  | 115.33 ± 15.37  | **1.67 ± 0.58** |
> |    | Normalized MSE   | 0.29295 ± 0.00008 | 0.29064 ± 0.00008 | **0.28575 ± 0.00013**|
> |    | Delta LM Loss (x1e-3) | 6.805 ± 0.074 | 6.694 ± 0.021  | **6.664 ± 0.066**|
> | 80 | Dead Feature     | 6311.33 ± 55.77  | 109.67 ± 11.50  | **1.00 ± 1.73** |
> |    | Normalized MSE   | 0.27787 ± 0.00003 | 0.27715 ± 0.00003 | **0.27341 ± 0.00003**|
> |    | Delta LM Loss (x1e-3) | 6.259 ± 0.005 | 6.217 ± 0.028  | **6.168 ± 0.033**|
>
> **We have incorporated these updates in Figure 5,6 and Appendix H of the revised manuscript.**

---

> ### Author Response · Authors · 2025-11-20
> **Statistical Significance Test (9 / 12)**
>
> > (Reviewer e5FR) Regarding random initialization: experiments should be repeated with different random seeds, with error bars shown and statistical significance reported.
>
> This is an important part we missed in our submitted manuscript and we are very thankful for pointing this out.
>
> To evaluate the performance differences between ASI and baseline methods (TopK and AuxK), we conducted **Welch's t-test** (also known as Welch's unequal variances t-test), a widely-used statistical hypothesis test that does not assume equal variances between the two groups being compared.
>
> ### Experimental Setup
> - Model, Layer, L0, Dictionary Size: Llama-3.1-8B, 15, 50, 32768
> -  Sample size : 15 independent runs for each method with different random seeds
> -  Metrics evaluated : Dead Feature Count, Normalized MSE and Delta LM Loss
> -  Objective : All metrics are "lower is better"
> -  Comparisons : ASI vs TopK and ASI vs AuxK across all three metrics
>
> For each comparison (ASI vs baseline method) on each metric, we formulated the following hypotheses:
>
> Null hypothesis (H₀) : $μ_{ASI}$ ≥ $μ_{baseline}$  (ASI performs worse than or equal to the baseline method on the given metric)
>
> Alternative hypothesis (H₁) : $μ_{ASI}$ < $μ_{baseline}$  (ASI performs better than the baseline method, i.e., achieves a lower metric value)
>
> This is a **one-tailed test** , as we specifically test whether ASI achieves significantly lower (better) values than the baseline methods.
>
> We used `scipy.stats.ttest_ind(ASI_values, baseline_values, equal_var=False, alternative='less')` to perform the Welch's t-test.
> The **p-value** represents the probability of observing the current data assuming the null hypothesis is true. A small p-value provides strong evidence against the null hypothesis:
>
> | Comparison     | Metric               | p-value        |
> |----------------|-------------------|----------------|
> | ASI vs TopK    | Dead Feature Count  | 3.26e-35       |
> | ASI vs TopK    | Normalized MSE      | 2.99e-40       |
> | ASI vs TopK    | Delta LM Loss       | 1.14e-23       |
> | ASI vs AuxK    | Dead Feature Count  | 4.33e-15       |
> | ASI vs AuxK    | Normalized MSE      | 6.33e-40       |
> | ASI vs AuxK    | Delta LM Loss       | **1.27e-06**       |
>
> **Given that the p-value is significantly smaller than any conventional significance level (0.05), we reject the null hypothesis. This indicates that ASI really outperforms the AuxK.**
>
> **We have incorporated these updates in Appendix I of the revised manuscript.**

---

> ### Author Response · Authors · 2025-11-20
> **Evaluate ASI across different hyperparameters (10 / 12)**
>
> > (Reviewer e5FR) Regarding hyperparameter selection: as, per Appendix G.5, it appears that authors have already experimented across batch size and learning rate. At minimum, I would suggest (a) reporting the results in these settings and indicating whether and why (not) the main-paper findings hold across these settings, and (b) additionally running experiments across dictionary size and K (and ideally alpha as well).
>
> Thanks for the reasonable and constructive suggestion.
>
> **We compare TopK, AuxK and ASI, using batch size 4096 and 32768**(learning rate is sweeped seperately). Other settings are the same(Llama-3.1-8B, layer 15, attention output, k=50, dictionary size=32768). The results are shown in the following table, which prove the main-paper findings hold across batch size.
>
> | Batch Size | Metric               | TopK       | AuxK      | ASI       |
> |------------|-------------------|------------|-----------|-----------|
> | 4096       | Dead Feature       | 16115      | 61        | **5**         |
> | 4096       | Normalized MSE     | 0.33351    | 0.32247   | **0.31685**   |
> | 4096       | Delta LM Loss (x1e-3) | 8.349  | 7.946     | **7.796**     |
> | 32768      | Dead Feature       | 21583      | 85        | **6**         |
> | 32768      | Normalized MSE     | 0.35377    | 0.34080   | **0.33033**   |
> | 32768      | Delta LM Loss (x1e-3) | 9.217  | 8.689     | **8.301**     |
>
> **We sweep the learning rate** for SAEs(L0=50, dictionary size=32768) trained on Llama-3.1-8B layer 15. Results in the following table.
>
> | Learning Rate | Normalized MSE | Dead Feature |
> |---------------|----------------|--------------|
> | 1e-5          | 0.33707        | 5373         |
> | 2e-5          | 0.33513        | 10511        |
> | 4e-5          | **0.33351**        | 16032        |
> | 8e-5          | 0.34163        | 20839        |
> | 1e-4          | 0.34575        | 21896        |
> | 2e-4          | 0.35863        | 23106        |
> | 4e-4          | 0.37526        | 24117        |
> | 8e-4          | 0.42229        | 26580        |
>
> Figure 6 shows the result of experiment **across K(L0)**. Figure 7 shows the result of experiment **across dictionary size(num of features)**. **These results prove the main-paper findings hold across these hyperparameters.**
>
>
> **We set alpha=1/32 following Gao et al.[1]**. Considering your concern, **we also sweep alpha for AuxK method**. The difference between used AuxK(alpha=1/32) and the optimal AuxK(alpha=1/4) is less than 4e-4, while the difference with ASI(MSE=0.31680) exceeds 3e-3. **This shows that the results are less sensitive to alpha in a reasonabble interval. The main-paper findings also hold.
>
> | Alpha       | Normalized MSE | Num of Dead Features |
> |-------------|----------------|--------------------|
> | 1/2         | 0.32207        | 16                 |
> | 1/4         | **0.32203**       | 19                 |
> | 1/8         | 0.32205        | 26                 |
> | 1/16        | 0.32221         | 42                 |
> | 1/32        | 0.32240        | 51                 |
> | 1/64        | 0.32275        | 65                 |
> | 1/128       | 0.32300        | 86                 |
> | 1/256       | 0.32347        | 87                 |
>
>
> [1] Leo Gao, Tom Dupré la Tou ... [Scaling and evaluating sparse autoencoders](https://arxiv.org/abs/2406.04093)

---

> ### Author Response · Authors · 2025-11-20
> **Evaluate ASI across different layers, models, datasets (11 / 12)**
>
> > (Reviewer e5FR) Model/layer/dataset: only a single model (Llama-3.1-8B), layer (15), and dataset (SlimPajama) are considered. All of the claims regarding the apparent superiority of ASI relative to baselines must be validated across multiple models, layers, and datasets. (In my opinion, given the setting, experimenting across a wider range of layers seems more important than doing so with respect to datasets, and datasets more important than models; but failing to consider multiple values for any of these experimental considerations is a clear red flag for cherrypicking.)
>
> Thanks for the reasonable and constructive suggestion.
>
> We further run experiments in the following settings:
>
> ## Dataset: SlimPajama; Model: Llama-3.1-8B; Activation Position: Layer 7 & 23 (attention output)
> | Layer | Metric             | Base      | AuxK      | ASI       |
> |-------|------------------|----------|----------|----------|
> | 7     | Dead              | 25836    | **98**       | 5308 |
> | 7     | Normalized MSE    | 0.32870  | 0.29800  | **0.28882** |
> | 7     | Delta LM loss (x1e-3) | 5.414 | 4.589   | **4.430** |
> | 23    | Dead              | 15542    | 332      | **27**    |
> | 23    | Normalized MSE    | 0.21302  | 0.20235  | **0.20141** |
> | 23    | Delta LM loss (x1e-3) | 1.942 | 1.865   | **1.845** |
>
> ## Dataset: fineweb-edu [1]; Model: Qwen3-8B; Activation Position: Layer 8 & 26 (attention output)
>
> | Layer | Metric             | Base      | AuxK      | ASI       |
> |-------|------------------|----------|----------|----------|
> | 8     | Dead              | 19048    | **4**    | 7        |
> | 8     | Normalized MSE    | 0.31228  | 0.28849  | **0.28533** |
> | 8     | Delta LM loss (x1e-3) | 2.956 | 2.591   | **2.566** |
> | 26    | Dead              | 16286    | **66**       | 566   |
> | 26    | Normalized MSE    | 0.30090  | 0.28127  | **0.28087** |
> | 26    | Delta LM loss (x1e-3) | 1.340 | 1.298   | **1.292** |
>
> Note: Llama-3.1-8B Layer 7 remains some dead features, it may because it has smaller effective rank(2351) compared to layer 23 (2654) and we don't specifically adjust d_init. Despite this, it still achieved the lowest MSE.
>
> **We have incorporated these updates in Appendix J of the revised manuscript.**
>
> [1] [https://huggingface.co/datasets/HuggingFaceFW/fineweb-edu](https://huggingface.co/datasets/HuggingFaceFW/fineweb-edu)

---

> ### Author Response · Authors · 2025-11-25
> **Clarification and Rewrite for  Section 2, 3 (12 / 12)**
>
> > (Reviewer e5FR) Background (sec 2):
> Discussion of linear subspaces (sec 2.1) focuses on the ill-defined notion of "low-rankness within self-attention mechanisms"; but there is in fact long line of work studying the phenomenon of linear subspace representation beyond attention mechanisms -- see, e.g., [1-5]. The subsequent discussion of the "linear representation hypothesis" in sec 2.2 focuses on the narrow interpretation of this hypothesis regarding 1-dimensional subspaces, when higher-dimensional subspaces (as studied in [1-5, inter alia]) would be a more relevant point of comparison for the intended study of intrinsic dimensionality.
>
> Thanks for your insightful feedback.
>
> **We added a dedicated paragraph in Section 2.1 of the revised manuscript summarizing the long-standing line of work on linear subspace representations and including all these related works.**
>
> > (Reviewer e5FR) The authors state that "the linear representation hypothesis... has been validated across diverse model scales, architectures, and modalities"; but the cited works do not (to my knowledge) conduct any specific hypothesis testing. Instead, they train SAEs across such contexts, and demonstrate their utility in interpretability; but the linear representation hypothesis remains very much in contention -- see, e.g., [5, 6].
>
> Thank you for your detailed feedback.
>
> **We have changed the statement to** ""Their successful application across a wide range of model scales, architectures, and modalities highlights their practical effectiveness for interpretability;  however, they do not constitute direct hypothesis tests of the superposition hypothesis, which remain active topics of debate" **in the Section 3.2  of the revised manuscript**.
>
> > (Reviewer e5FR) The authors state that "This formulation shows that O can be viewed as the sum of low-dimensional outputs from each head." However, it shows only that there exists a linear transformation from Z to O, which may be full-rank. (Note that the authors use "rank" and "dimension" interchangeably; so I am interpreting the use of "low-dimensional" in the context of a linear transformation to mean "low-rank" -- please correct this interpretation if I am wrong.)
>
> Sorry, our formula here is not clear enough. We further elaborated the calculation expression for O as follows:
>
> Let $Z = \text{Concat}[Z_1, \dots, Z_H] \in \mathbb{R}^{n \times d}$ denote the concatenated outputs of all attention heads.
> The final attention output is obtained by applying the output projection:
>
> $
> O = Z W^O = \sum_{i=1}^{H} Z_i W^O_i,
> $
>
> where each $W^O_h \in \mathbb{R}^{d_h \times d}$ is the corresponding submatrix of $W^O \in \mathbb{R}^{d \times d}$ associated with head $i$.
> $Z_i W^O_i$ represents the low-dimensional outputs from each head.
>
> **We have updated this part in Section 3.1 of the revised manuscript**.

---

### Author Response · Authors · 2025-12-02
**Overall Summary of Reviews and Key Revisions**

We sincerely thank all reviewers for their thoughtful feedback and constructive suggestions. Below, we summarize the key areas where reviewers expressed **positive recognition**:
- Our effective dimensionality probing was recognized as **principled** (*reviewer GERd*), the resulting finding that attention outputs lie in a lower-dimensional subspace was viewed as **clear** and **convincingly demonstrated** (*reviewers fThX and TjHM*), and our analysis of the source of this low-rank structure was noted as a **valuable follow-up** (*reviewer TjHM*).
- Our finding that the low-dimensional structure of attention outputs is linked to the emergence of dead SAE features was recognized as **significant** (*reviewer e5FR*) and highlighted as an **insightful**, **well-supported** contribution (*reviewers GERd and fThX*).
- The proposed initialization strategy was recognized as **well-motivated** and **practically effective** for mitigating dead features (*reviewers e5FR, GERd, and fThX*) and was viewed as a **novel** idea with potential broader benefits for SAE training efficiency (*reviewer TjHM*).

We are particularly thankful for the reviewers’ insightful suggestions, which has been instrumental in refining our work. In summary, our rebuttal has addressed the core concerns regarding **Writing**, **Empirical Rigor**, **Statistical Soundness**, **Generalizability** and **Potential Risks**:
- **Writing**: We addressed terminology issues by replacing the nonstandard “intrinsic dimension” with the standardized “effective rank” and recalculated relevant results, which do not affect our conclusions [*Response2 to Reviewer e5FR*]. Content and figures were clarified and partially rewrited[*Response3,4,5,6 to Reviewer e5FR; Response2,3,4,5 to Reviewer GERd; Response6 to Reviewer TjHM*]. We also added relevant references and related work discussions[*Response12 to Reviewer e5FR*].
- **Empirical Rigor**: For our finding "low effective dimension correlates with dead features", we verified that it holds across activation types, layers and SAE hyperparamters[*Reponse7 to Reviewer e5FR*]. We evaluated our proposed initialization method under different hyperparameters, demonstrating its superiority[*Reponse10 to Reviewer e5FR*].
- **Statistical Soundness**: We calculated the mean and standard deviation for the core experimental results[*Response8 to Reviewer e5FR*] and performed statistical significance tests to ensure that the experimental results were not affected by random noise[*Response9 to Reviewer e5FR*].
- **Generalizability**: We validated the effectiveness of our method across different activation type[*Response2 to Reviewer fThX*], models, layers, and datasets[*Response11 to Reviewer e5FR*]. We calculated the effective dimension for the 5 types of activations commonly used in training SAEs in the model[*Response5 to Reviewer TjHM*].
- **Potential Risks**: We addressed concerns about whether our method could compromise interpretability, miss features in the dead subspace, or induce feature-splitting. Additional analyses (monosemanticity evaluation[*Response4 to Reviewer fThX*], feature coverage of dead subspace, reconstruction of dead subspace[*Response2 to Reviewer TjHM*], and scaling-law clarification[*Response3 to Reviewer TjHM*]) show that our method does not introduce these risks.

We would like to express our gratitude to the reviewers, the originally assigned AC, and the newly assigned AC for their time and thoughtful engagement throughout the review process. To assist the newly assigned AC quickly get an overview of the situation, we provide a concise summary of each reviewer’s main concerns, our rebuttals, and the reviewers’ follow-up responses in the following.

---

> ### Author Response · Authors · 2025-12-02
> **Summary of Reviewer Concerns, Rebuttals, and Follow-Up Status**
>
> ## Reviewer e5FR Score 2 Conf 3 (No response after rebuttal)
> The Reviewer e5FR raised the following questions regarding **Writing**, **Empirical Rigor**, **Statistical Soundness** and **Generalizability**:
> > This definition of "intrinsic dimension" is nonstandard
>
> We replaced the term with the standardized “effective rank” and confirmed this correction does not alter our conclusions.
> > In many cases, there is not enough technical information regarding what experiments have been performed to properly interpret results
>
> We have added a summary of the settings and updated all unclear contents, figures and captions.
> > The authors claim that "low intrinsic dimensionality strongly correlates with the number of dead features". However, there is no systematic analysis of obvious confounders such as dictionary size or the setting of K (L0) with which (top-K) SAEs are trained.
>
> We have trained and included results across different dictionary sizes and L0.
> > experiments should be repeated with different random seeds, with error bars shown and statistical significance reported.
>
> We have used 3 random seeds to compute the errors and 16 random seeds to conduct Welch's t-test.
> > All of the claims regarding the apparent superiority of ASI relative to baselines must be validated across multiple models, layers, and datasets.
>
> We have validated across different models, layers, datasets, dictionary sizes and L0.
>
> ---
> ## Reviewer GERd Score 4 Conf 3 (No response after rebuttal)
>
> The Reviewer GERd raised just one weakness regarding **Writing**:
> > The biggest issue I have with the paper is that the experimental section seems to be disorganized. It is hard to understand which models, datasets, and setups were used to prepare the experiments and support the claims.
>
> We have added a summary of the settings and updated all unclear contents, figures and captions.
>
> ---
> ## Reviewer fThX Score 6 Conf 4 (No response after rebuttal)
>
> The Reviewer fThX raised 3 questions regarding **Generalizability** and **Potential Risks**:
> > In Figure 4 the typical effective rank for MLPs is about 0.90–0.95, but the last layer shows a drop to roughly 65%. Any insights into what might cause this behavior? Did you apply your initialization to this layer as well, given it also shows a nontrivial fraction of dead features?
>
> We verified this phenomenon on different models and confirmed that our method remains effective in this case.
> > Architectural implications of low-rank attention: If attention outputs are low rank, could one reduce per-head dimensionality while keeping d_model fixed, possibly increasing the number of heads, without losing much information?
> > Is the observed low rank due to redundancy across heads or primarily to W_O anisotropy?
>
> We offer our thoughts on these discussions, but these are beyond the scope of this article; we leave them for future work.
> > The paper does not discuss whether the initialization helps interpretability. Even a brief note or example on whether feature quality or concept alignment improves would be useful, though I understand this is not the main focus.
>
> We have evaluated the monosemanticity of the features and found our initialization methods have no impact.
>
> ---
> ## Reviewer TjHM Score **4**=>**6** Conf 4 (Reply on November 25th)
> The Reviewer TjHM raised several concerns regarding **Writing**, **Generalizability** and **Potential Risks**:
> > SAEs trained with this initialization might end up missing certain features. Even though the active subspace contains the vast majority of the variance (99%), SAEs have many features and one would expect that roughly 1% of them should fall into the “dead” subspace.
>
> We demonstrate that our method has no impact on performance in dead subspaces by examining both feature coverage and reconstruction behavior.
> > Figure 7(b) is confusing, are they saying that their SAEs find up to 1M alive features in an 8B model? I would expect insanely high levels of feature splitting at that point
>
> We clarified that our method aims to prevent dead features at SAE sizes where they should not naturally arise. This result is only to demonstrate its effectiveness across a range of SAE sizes.
> > As far as I’m aware, training SAEs on the attention output is somewhat uncommon, so if that is the only activation space where this technique is relevant, most real-world use cases would not benefit from this approach.
>
> We explained that our initialization method can apply in domain-specific datasets, which has low-rank residual streams.
> > The authors do not study other activation spaces which are commonly used when training SAEs
>
> We further compute 5 types of activations commonly used in SAE training.
> > Many of the figures are difficult to interpret.
> > The definition of the initialization method itself is somewhat unclear.
>
> We have added a summary of the settings and updated all unclear contents, figures and captions.
>
> **According to our rebuttal, the Reviewer TjHM raised the score from 4 to 6 on November 25th.**

---

### Meta-Review · Area_Chair_j1ba · 2026-01-05

**Summary:**

The initial reviews are largely in favor of rejection, with only one reviewer voting for acceptance (and then only marginally so).

Concerns raised by the reviewers are that the authors' definition of "intrinsic dimension" (a core component of their work) is somewhat ad-hoc and non-standard in the literature, issues with the experimental evaluation and conclusions drawn from experiments, questions regarding how relevant the work will be to alternative architectures, and unclear presentation / lack of sufficient technical detail in many areas.

In their rebuttal the authors have made a significant effort to address a number of these concerns with additional experiments, computing alternative metrics, and clarifying points of the presentation.  However, the scope of the required changes appears significant and, in my opinion, is still unlikely to change the reviewers' scores sufficiently to reach a consensus to accept.  Were this a journal submission, it would appear to be a clear-cut major revision and re-review decision, but as this is unfortunately not possible I recommend that the paper be rejected and encourage the authors to submit a revised version to future venues.

**Reviewer Concerns:**

The authors have made a very strong effort to address the reviewers' concerns, with one reviewer noting that the majority of their concerns were addressed and raising their score (though that reviewer still notes unresolved concerns).  However, overall, it would appear to me that several issues remain outstanding.

In particular, reviewer e5FR raises numerous technical concerns with the work.  While the authors have attempted to address many of these, to do so has required going so far as to redefine concepts and metrics core to their work (e.g., changing their definition of Intrinsic Dimension to Effective Rank).  The authors argue that this change does not effect their overall results, but it seems quite the jump to assume the reviewer would be convinced by this argument, particularly given the numerous other technical and clarity concerns raised.

Likewise, reviewer GERd raises numerous concerns with the presentation and organization of the experimental results (concerns also echoed by other reviewers).  While many of these questions are clarified by the authors in the rebuttal.  Overall it still tends to speak to the level of the revision that is required in the manuscript.

Reviewer TjHM explicitly notes that most of their concerns have been address except for a comment that the results appear to be rather limited in what architectures they can be applied to (namely, Sparse AutoEncoders).

Reviewer fThX was the most positive in their initial review, and most of their questions appear to be answered by the authors.

**Reviewer Scores:**

Overall, while one reviewer explicitly notes increasing their score, it is very questionable to me that the two remaining critical reviews (e5FR and GERd) would increase their scores sufficiently to argue for acceptance.   Moreover, the scope of the required changes to the manuscript appear significant, and it would seem the work would be better served by submitting a major revision to a future conference/journal to incorporate the significant feedback from the reviewers.

---

### Decision · Program_Chairs · 2026-01-26

Reject